Methods

# Estimation of crossbridge-state during cardiomyocyte beating using second harmonic generation

Hideaki Fujita[1], Junichi Kaneshiro[2], Maki Takeda[3], Kensuke Sasaki[2], Rikako Yamamoto[1], Daiki Umetsu[6,7], Erina Kuranaga[6], Shuichiro Higo[4] , Takumi Kondo[4], Yoshihiro Asano[5], Yasushi Sakata[5], Shigeru Miyagawa[3], Tomonobu M Watanabe[1,2]

**Estimation of dynamic change of crossbridge formation in living cardiomyocytes is expected to provide crucial information for elucidating cardiomyopathy mechanisms, efficacy of an intervention, and others. Here, we established an assay system to dynamically measure second harmonic generation (SHG) anisotropy derived from myosin filaments depended on their crossbridge status in pulsating cardiomyocytes. Experiments utilizing an inheritable mutation that induces excessive myosin–actin interactions revealed that the correlation between sarcomere length and SHG anisotropy represents crossbridge formation ratio during pulsation. Furthermore, the present method found that ultraviolet irradiation induced an increased population of attached crossbridges that lost the force-generating ability upon myocardial differentiation. Taking an advantage of infrared two-photon excitation in SHG microscopy, myocardial dysfunction could be intravitally evaluated in a *Drosophila* disease model. Thus, we successfully demonstrated the applicability and effectiveness of the present method to evaluate the actomyosin activity of a drug or genetic defect on cardiomyocytes. Because genomic inspection alone may not catch the risk of cardiomyopathy in some cases, our study demonstrated herein would be of help in the risk assessment of future heart failure.**

## Introduction

Cardiomyopathy, a disease of cardiac dysfunction, is the worldwide leading cause of sudden death in young people, including children (Rizzo et al, 2019). Muscle fiber contractions in the heart are synchronized through cell adhesion and electrical transmission during normal beating. Abnormalities in any of the elements of an integrated heart system may result in heart failure (Richardson et al, 1996; Caforio et al, 2013). Among them, muscle contraction, the driving force of the heartbeat, is a promising target for understanding the occurrence mechanism and treatment of systolic heart failure (Walsh et al, 2010; Moore et al, 2012). In cardiomyocytes, cardiac myosin, a cytoskeletal motor protein, generates contractile forces along an actin filament using energy stored in adenosine triphosphate (ATP) under the control of various regulatory proteins (Hanft et al, 2008; Barrick & Greenberg, 2021). Muscle force generation dysfunction in the heart is mainly caused not by mutation in the myosin motor itself but mutations in myosin regulatory/accessory proteins via malfunctions of various pathways including signal transduction, calcium ($Ca^{2+}$) cycling, adenine nucleotide transportation, reactive oxygen species production, and so on (Richardson et al, 1996; Caforio et al, 2013). To reveal the causal relationship from mutation to heart failure, and to investigate the toxicity, the severity, or the efficacy of a drug against cardiomyocytes, it is necessary to quantify their effects on actomyosin activity during muscle contraction in living cardiomyocytes.

Heartbeat measurement based on video analysis is a simple noninvasive method for evaluating cardiomyocyte dysfunction (Maddah et al, 2015; Santoso et al, 2020). However, it is not applicable for selectively focusing on myosin motor function. Changes in the length of the sarcomere, the smallest functional unit of the muscle, is an optically visible dynamic indicator of muscle function, but is still not sufficient to evaluate actomyosin activity. It would be ideal to directly measure the myosin force generation in living cardiomyocytes. However, various techniques for measuring the forces exerted by myosin molecules, myofibers, and myocytes, developed from a long history of muscle mechanobiology, are almost all contact measurements, such as atomic force microscopy, traction microscopy, and magnet/laser trapping, or invasive measurements, such as laser ablation (Roca-Cusachs et al, 2017; Woody et al, 2018). This is because force is defined through the properties of a material and its physical deformation. A noninvasive/noncontact method for measuring actomyosin activity in living muscle cells, even without directly measuring force itself, could accelerate elucidation of the mechanism and drug discovery for

[1]Department of Stem Cell Biology, Research Institute for Radiation Biology and Medicine, Hiroshima University, Hiroshima, Japan   [2]Laboratory for Comprehensive Bioimaging, RIKEN Center for Biosystems Dynamics Research, Kobe, Japan   [3]Department of Cardiovascular Surgery, Osaka University Graduate School of Medicine, Osaka, Japan   [4]Department of Medical Therapeutics for Heart Failure, Osaka University Graduate School of Medicine, Osaka, Japan   [5]Department of Cardiovascular Medicine, Osaka University Graduate School of Medicine, Osaka, Japan   [6]Laboratory for Histogenetic Dynamics, Graduate School of Life Sciences, Tohoku University, Sendai, Japan   [7]Department of Biological Sciences, Graduate School of Science, Osaka University, Osaka, Japan

Correspondence: tomowatanabe@riken.jp

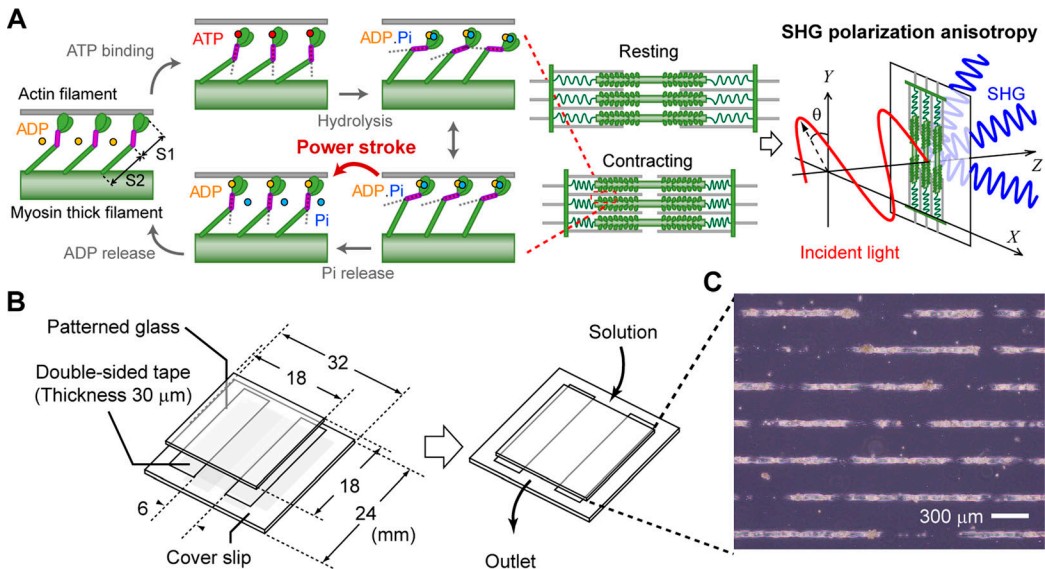

**Figure 1. Schematic illustration showing the concept of this study.**
**(A)** Explanative drawing of the concept of probing actomyosin conformation in a sarcomere by SHG anisotropy. Coupling with ATP hydrolysis, myosin changes conformation of its S1 region causing angular change of its S2 fragment, which is called mechanochemical coupling. The detected SHG anisotropy was altered by the attachment/detachment of myosin S1 to/from actin. **(B)** Explanative drawing of our sample preparation. **(C)** A microscopic photo of cardiomyocytes adhered on a line-and-space pattern substrate.

cardiomyopathy and the research in mechanobiology. In this study, we focused on the structural dynamics of the actomyosin complex during force generation. The structural state of actomyosin in a sarcomere determines the anisotropic features of optical second harmonic generation (SHG) (Plotnikov et al, 2006; Nucciotti et al, 2010; Schurmann et al, 2010; Psilodimitrakopoulos et al, 2014; Forderer et al, 2016; Yuan et al, 2019). This indicates that SHG polarization microscopy is a potential alternative to direct force measurements in muscles.

SHG is a nonlinear coherent scattering process that reflects permanent electric dipole moments and their alignments in illuminated materials (James & Campagnola, 2021). The SHG light is strongly generated from fibrous materials with asymmetries in electric polarization, such as fibrillar collagen in tendons, myosin thick filaments in myosarcoma, and densely bundled microtubules (Mohler et al, 2003; Cox, 2011; Cicchi et al, 2013). The coefficients representing SHG, a third-rank polar tensor, originate from the electric dipole moments and can be measured using the polarization dependence of SHG (James & Campagnola, 2021). Although both myosin and actin in the actomyosin complex emit SHG, the SHG from myosin is over three orders of magnitude larger than that of actin (Plotnikov et al, 2006). Myosin is composed of a tadpole-shaped subdomain-1 (S1), an α-helical coiled-coil subfragment-2 (S2), and a light meromyosin (LMM) region for rod filamentation (Fig 1A, left). The S1 region converts chemical energy from ATP hydrolysis into mechanical energy by changing its structure corresponding to the nucleotide state, called "lever arm swinging." The LMM is responsible for rod filamentation, whereas the S2 region acts as a "spring" between the converter and the rod (Barrick & Greenberg, 2021). The primary source of SHG in muscle is thought to be the S2 and the LMM, whose structures comprise highly regular

unidirectional filaments (Plotnikov et al, 2006). In contrast, the contribution of the globular S1 to SHG is limited to approximately 10% (Nucciotti et al, 2010). During muscle contraction, S1 is dynamic, LMM is static, and S2 interlocks with S1 dynamics. Therefore, the tilting or bending of the S2 region coupled with myosin force generation alters SHG anisotropy in a contracting sarcomere, which can be detected through the incident polarization dependence of SHG intensity (Fig 1A, right). Several groups have reported their success in discriminating SHG anisotropy between rigor and relaxed states in rabbit psoas fibers (Nucciotti et al, 2010), isolated scallop myofibrils (Plotnikov et al, 2006), mouse tibialis anterior (Schurmann et al, 2010), and mouse-skinned extensor digitorum longus muscle fibers (Forderer et al, 2016). In addition, the dynamic measurement of SHG anisotropy has been achieved in a dissected intact frog tibialis anterior muscle fiber during isometric force generation (Nucciotti et al, 2010), worm walking in *Caenorhabditis elegans* (Psilodimitrakopoulos et al, 2014), and collagenase-treated interossei cells (Forderer et al, 2016). The dependence of SHG anisotropy on longitudinal and lateral stretch stresses in cardiomyocytes was measured in a relaxed state in a chemically fixed sliced sample (Yuan et al, 2019).

To the best of our knowledge, there have been no reports on measuring SHG anisotropy in beating single cardiomyocytes. In this study, we constructed an assay system to measure dynamic changes in SHG anisotropy emitted from a sarcomere in a living cardiomyocyte using a highly sensitive SHG polarization microscope with a fast polarization-controllable device, as previously described (Kaneshiro et al, 2016; Shima et al, 2018). The key gadget was a line-and-space pattern substrate, which could solve the problem of sarcomere arrangement irregularity, making the experiment difficult. Combining high-speed SHG polarization microscope

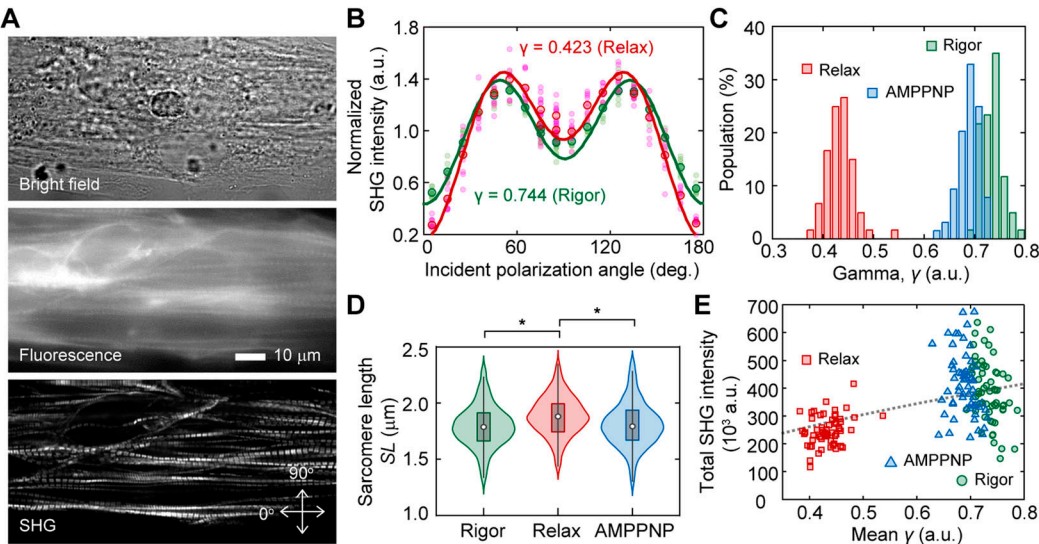

**Figure 2.  Probing structural conformation of actomyosin complex by SHG anisotropy.**
**(A)** Typical example of SHG image acquisition of cardiomyocytes on a patterned line. Top, bright field image; middle, fluorescent image stained with fluorescent phalloidin; bottom, SHG image. Arrows indicate the orthogonal axes of the incident polarization. **(B)** Incident polarization dependence of SHG emitted from a sarcomere in the absence (*green*, *Rigor*) or the presence of ATP (*red*, *Relax*). Pink circles and light green circles are typical 10 traces in the absence or the presence of ATP, respectively. Red circles and green circles are mean value of the 10 traces of them. Solid lines are fitting results with Equation (2). The value of SHG intensity is normalized by the intensity of averaged value of all obtained data for relax state at 90°. **(C)** Histograms of mean value of parameter γ in a field of view (FOV) in the absence (*green*, N = 60 FOVs) or the presence of ATP (*red*, N = 64 FOVs), and in the presence of AMPPNP (*blue*, N = 64 FOVs). **(D)** Violin plots of sarcomere length in the absence (*green*, N = 1,339 sarcomeres) or the presence of ATP (*red*, N = 1,420), and in the presence of AMPPNP (*blue*, N = 1,486). Asterisks indicate less than 0.01 of the *P*-value in the *t* test. **(E)** Correlation between the mean γ and the total SHG intensity in a FOV in the absence (*green*, N = 60 FOVs) or the presence of ATP (*red*, N = 60 FOVs), and in the presence of AMPPNP (*blue*, N = 64 FOVs). The black broken line is the theoretical curve based on the integration of Equation (2).

and line-and-space pattern substrate enabled us to perform SHG anisotropy measurement in beating cardiomyocyte. Here, we investigated the relationship between SHG polarization originating from actomyosin activity and disease phenotypes in two models: one based on induced pluripotent stem cell (iPSC) technology of genetic hypertrophic cardiomyopathy and another on acquired muscle failure induced by ultraviolet irradiation. Because SHG measurement is based on two-photon excitation optics, the wavelength of the irradiated light is in the infrared region, enabling deep tissue imaging. As a more challenging experiment, we further applied the present method to evaluate actomyosin dysfunction in a *Drosophila* model of Barth syndrome (Xu et al, 2006). The present actomyosin evaluation method based on the dynamic measurement of SHG anisotropy developed in this study promises to provide a new effective tool in cardiomyopathy research and medicine.

## Results

### Construction of an assay system for SHG polarization measurement in cardiomyocytes

SHG is a nonlinear scattering phenomenon that occurs in both the forward and backward directions. In general, SHG intensity is stronger during forward scattering than backward scattering (James & Campagnola, 2021), 26-fold higher in our study (Fig S1).

Forward- and backward-scattered SHGs have different implications; because the backward-scattered SHG contains signals derived from multiple scattering of the forward-scattered SHG when observing a thick sample, the backward scattered SHG is difficult to interpret (Williams et al, 2005; Chu et al, 2009). Meanwhile, measuring the backscattered SHG is suitable for biological sample observations. Here, we chose a transmission SHG microscope because of its high signal strength for fast measurements during a cardiomyocyte pulsation (Fig S2) and simple interpretation because of the absence of signal contamination. In a transmission-type microscope, the sample should be placed between two objectives. A special chamber was composed of two glass plates and a polyester spacer with dimensions of 18 mm × 6 mm × 0.03 mm (Fig 1B). A line-and-space pattern substrate allowed cardiomyocytes to attach in a linear fashion (Fig 1C).

Here, we prepared spontaneously beating cardiomyocytes differentiated from human iPSCs (hiPSCs), 253G1 strain. The sarcomeres in chemically permeabilized cardiomyocytes were arranged along the employed line-and-space pattern and selectively visualized via SHG (Fig 2A). The state transition of the actomyosin crossbridge from rigor to relaxation in permeabilized cardiomyocytes could be induced by loading an ATP solution into the chamber on the microscope. Following the experimental procedure in previous reports (Plotnikov et al, 2006; Nucciotti et al, 2010; Schurmann et al, 2010), we measured the SHG intensity at various incident polarizations of a sarcomere with (in relax state) or without (in rigor state) 5 mM ATP (Fig 2B). Polarization dependence was expressed

using the following equation, assuming the dipole polarity angle $\varphi$ of the summation of the S1 and S2 directions within a confocal volume:

$$I(\theta) = \{a\cos^2(\theta - \theta_0) + b\sin^2(\theta - \theta_0)\}^2 + \sin^2 2(\theta - \theta_0) \qquad (1)$$

$$a = \beta_0 \cos^3\varphi, \quad b = \frac{1}{2}\beta_0 \cos\varphi\sin^2\varphi$$

where $\theta$ is the incident polarization, $\theta_0$ is the orientation angle of the sarcomere in microscope coordinates, and $\beta_0$ is a proportional constant (Tiaho et al, 2007). The parameter $\gamma$, which reflects the average crossbridge state (Plotnikov et al, 2006; Nucciotti et al, 2010; Schurmann et al, 2010), was estimated by fitting the obtained polarization dependency of the intensity using the following equation:

$$I(\theta) = A\left[\left\{\gamma\cos^2(\theta - \theta_0) + \sin^2(\theta - \theta_0)\right\}^2 + \sin^2 2(\theta - \theta_0)\right] \qquad (2)$$

where $A$ is the proportionality constant (Fig 2B, *solid lines*).

The mean value of $\gamma$ in a field of view (FOV) of 128 × 512 pixels (25 × 100 $\mu$m) clearly decreased from $\gamma_{rigor} = 0.73 \pm 0.02$ (mean ± SD) in the rigor state to $\gamma_{relax} = 0.43 \pm 0.03$ in the relaxed state (Fig 2C, *red and green*). These values were consistent with the values reported in previous studies, where $\gamma_{rigor} = 0.68 \pm 0.01$ and $\gamma_{relax} = 0.46 \pm 0.03$ for rabbit psoas fibers (Nucciotti et al, 2010), $\gamma_{rigor} = 0.73 \pm 0.04$ and $\gamma_{relax} = 0.50 \pm 0.05$ for mouse tibialis anterior (Schurmann et al, 2010), and $\gamma_{rigor} = 0.70 \pm 0.06$ and $\gamma_{relax} = 0.45 \pm 0.04$ for mouse extensor digitorum longus fibers (Forderer et al, 2016). Loading 5 mM of 5′-adenylylimidodiphosphate (AMPPNP), a non-hydrolyzable analog of ATP, slightly but significantly decreased $\gamma$ to $\gamma_{AMPPNP} = 0.69 \pm 0.02$ ($P = 5.3 \times 10^{-22}$ in $t$ test) (Fig 2C, *blue*). Although AMPPNP was thought to induce myosin dissociation from actin, sarcomere length ($SL$) was not increased by AMPPNP treatment but was increased by ATP ($P = 1.3 \times 10^{-37}$ in $t$ test) (Fig 2D).

According to Equation (2), the integral of the obtained SHG intensity should monotonously increase to the $\gamma$-value. The total SHG intensity in the FOV decreased after the decrease in $\gamma$ after treatment with ATP, but not with AMPPNP (Fig 2E). Although a monotonous increase was observed within the ATP data (Fig 2E, *red*), the intensity was negatively correlated with the $\gamma$-value within the rigor or AMPPNP data (Fig 2E, *green and blue*), indicating the presence of a factor contributing to intensity but not related to the $\gamma$-value. Thus, SHG anisotropy measurement, apart from intensity measurement, could be meaningful for probing actomyosin activity.

## Dynamic measurement of SHG anisotropy in living cardiomyocytes

The SHG polarization microscope we constructed enables the measurement of the incident polarization dependence of SHG within 1 ms/pixel (Kaneshiro et al, 2016; Kaneshiro et al, 2019). By limiting the FOV to 2 × 40 pixels (0.39 × 7.81 $\mu$m), including approximately 2–3 sarcomeres, measurement within 80 ms can be possible (Fig 3A). The dynamic spike-like change in $\gamma$ synchronized with sarcomere contraction was successfully measured during spontaneous beating in living cardiomyocytes derived from hiPSCs (Fig 3B). The average $\gamma$ at the resting state $\gamma_{rest}$ and the contracting

state $\gamma_{cont}$ could be estimated by fitting the histogram of $\gamma$ with a double Gaussian distribution (Fig 3C). The $\gamma_{rest} = 0.43$ obtained corresponded to the $\gamma_{relax}$ (0.43) obtained in permeabilized cardiomyocytes, whereas the $\gamma_{cont} = 0.50$ was smaller than $\gamma_{rigor}$ (0.69). This tendency was the same as that previously reported for isometric force generation in frog muscle fibers (Nucciotti et al, 2010). The value of $\gamma_{cont} = 0.50$ was reasonable, assuming that $\gamma$ reflects the population of myosin bound to actin. The value of $\gamma_{cont}$ can be predicted using the obtained $\gamma_{rigor}$ and $\gamma_{relax}$, (0.69 – 0.43) × 0.25 + 0.43 = 0.495, because the population of myosin contributing to force generation during contraction is only between 20–30% (Piazzesi et al, 2007). During the continuous beating cycle, the relationship between sarcomere length $SL$ and $\gamma$ followed approximately the same trajectory (Fig 3D), indicating a one-to-one relationship between sarcomere motion and $\gamma$ dynamics.

Plots of the average $SL$ and $\gamma$ trace during pulsation highlighted that local sarcomere shortening preceded the increase in $\gamma$, whereas the decrease in $\gamma$ preceded sarcomere lengthening (Fig 3E, *top and middle*), indicating that the relationship between $SL$ and $\gamma$ was nonlinear during the transition process. Average sarcomere lengths in the contracting and resting states, denoted as $SL_{cont}$ and $SL_{rest}$, respectively, were measured using the average SHG intensity profiles during the contracting and resting states with reference to the $\gamma$-value (Figs 3F and S3): the mean $SL_{rest}$ and $SL_{cont}$ were determined to be 1.94 ± 0.10 $\mu$m and 1.77 ± 0.10 $\mu$m (N = 25 FOVs), respectively, which were consistent with commonly known values measured in myocardial cells (Hanft et al, 2008). The correlations of $\gamma$–$SL$ in a 40-s time course were plotted on the same line, regardless of being the contracting or resting state (Fig 3G). Instead of the $SL$ measurement, the change in sarcomere shape from the relaxed state could be also quantified by the correlation between the SHG images in the resting and contracting states (Fig 3B, *bottom*, and 2E, *bottom*). This method lost the length information and the synchronization timing between local sarcomere contraction and the $\gamma$-value because the motion of sarcomeres in a FOV was affected by those outside the FOV. Nevertheless, the contraction duration and relaxation delay during sarcomere motion can be obtained without care of defocusing because of muscle contraction.

By treating cardiomyocytes with 10 $\mu$M blebbistatin (BS), a widely used inhibitor of actomyosin ATP hydrolysis (Straight et al, 2003; Kampourakis et al, 2018), pulsation was stopped because of the inactivation of myosin force generation (Fig 4A, *upper*). BS is known to retain the myosin ATPase cycle in the ADP.$P_i$ state by inhibiting the release of phosphate (Kovacs et al, 2004). The average $\gamma$ in a 40-s time course, $\gamma_{ave}$, was increased after treatment with 10 $\mu$M BS from 0.41 ± 0.09 without BS to 0.48 ± 0.10 ($P = 4.7 \times 10^{-15}$ in $t$ test) (Fig 4C, *red and magenta*). By inactivating the BS effect via blue laser irradiation (Kolega, 2004; Sakamoto et al, 2005), cardiomyocytes resumed beating, and the time course of $\gamma$ exhibited periodic spiking, followed by a decrease in $\gamma$ during the resting state (Fig 4A, *lower panel*). Because not all sarcomeres showed recovery of pulsation, no significant difference was observed in the histogram of $\gamma_{ave}$ before and after the irradiation, but a small decrease was confirmed in the mean value of $\gamma_{ave}$, 0.47 ± 0.11 (Fig 4C, *magenta and blue*). Treating cardiomyocytes with 10 $\mu$M omecamtiv mecarbil (OM), a cardiac-specific myosin activator and a potential pharmaceutical cure for heart dysfunction (Teerlink, 2009; Malik et al,

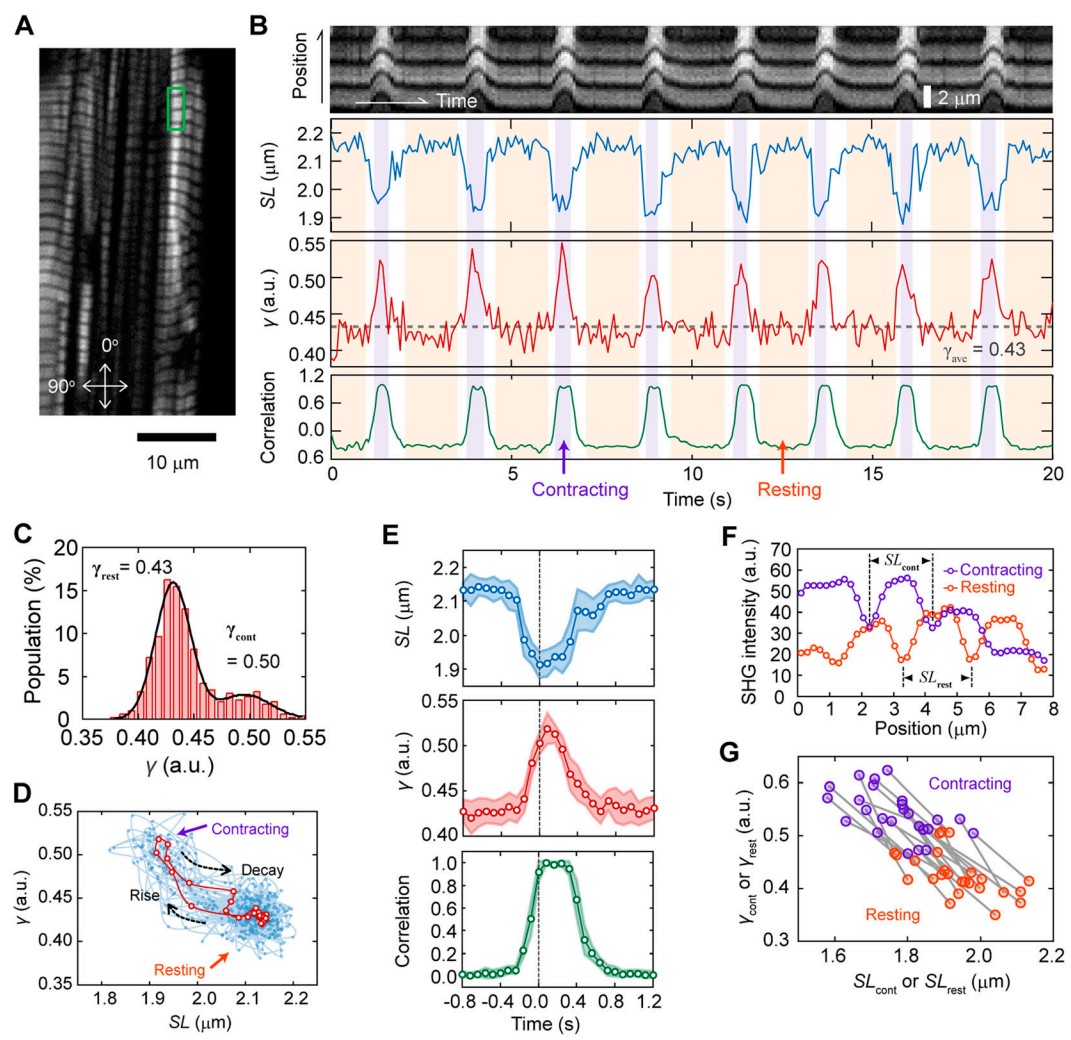

**Figure 3. Probing dynamic structural conformation of actomyosin complex in a living cardiomyocyte.**
**(A)** A typical example of SHG image acquisition of a living cardiomyocytes. Arrows indicate the orthogonal axes of the incident polarization. **(B)** A typical example of dynamic measurement of parameter γ with 80 ms time resolution at the area shown in green rectangle in (A). Top, the image kymograph based on the total SHG intensity; second top, sarcomere length SL; second bottom, parameter γ; bottom, sarcomere motion converted into numerals by calculating correlation with an image at relaxation state. Light magenta and orange in the back indicates contracting and resting phases, respectively. **(C)** Histogram of γ within a trace of 40 s. Black line is a fitting result with double Gaussian distributions. **(D)** Relationship between SL and γ obtained in data shown in (B). Cyan circles and lines are all raw data, and red ones are the average values shown in E, *middle*. **(E)** Average traces of SL (*top*), γ (*middle*), and sarcomere motion (correlation) (*bottom*) in 16 pulsations for 40 s. The time zero was adjusted at the time of the largest value of SL. Light colors are SD. **(F)** Cross-sections of the SHG image averaged in the contraction state (*magenta*) or resting state (*orange*). **(G)** Correlation plots between $SL_{rest}$ or $SL_{cont}$ and $γ_{rest}$ or $γ_{cont}$ of 25 traces. The pair of $SL_{rest}$–$γ_{rest}$ and $SL_{cont}$–$γ_{cont}$ obtained from a trace is linked by a gray line.

2011; Kampourakis et al, 2018), remarkably reduced the occurrence of pulsation, making it difficult to capture pulsation within 40 s of measurement (Fig 4B). Unlike BS, which is known to interact with the nucleotide-binding pocket in S1, OM is known to bind to an allosteric site to stabilize the lever arm in a position before force generation, resulting in a greater population of myosin cross-bridges without changing the structure of S1 (Winkelmann et al, 2015; Planelles-Herrero et al, 2017; Woody et al, 2018). In view of this inhibitory mechanism, OM treatment was expected to increase $γ_{ave}$ above $γ_{cont}$ according to the previously proposed interpretation that γ reflects the crossbridge population. Indeed, the mean $γ_{ave}$ was increased to 0.52 ± 0.11 ($P$ = 1.1 × 10$^{-23}$ in $t$ test) upon treatment with 10 μM OM (Fig 4C, *green*). The increase in $γ_{ave}$

after OM treatment was significantly larger than that by BS, both before and after inactivation ($P$ = 0.007 and 0.001, respectively, in $t$ test). In summary, the dynamic measurement of SHG polarization anisotropy provided quantitative parameters to assess actomyosin crossbridge formation in living cardiomyocytes.

## Noninvasive evaluation of force generation dysfunction and repair in cardiomyocytes derived from hiPSC obtained from patients with genetic heart disease

We further elucidate the relationship between crossbridge formation and SHG anisotropy in sarcomeres as a demonstration of the applicability of the present method to evaluate a dysfunctional

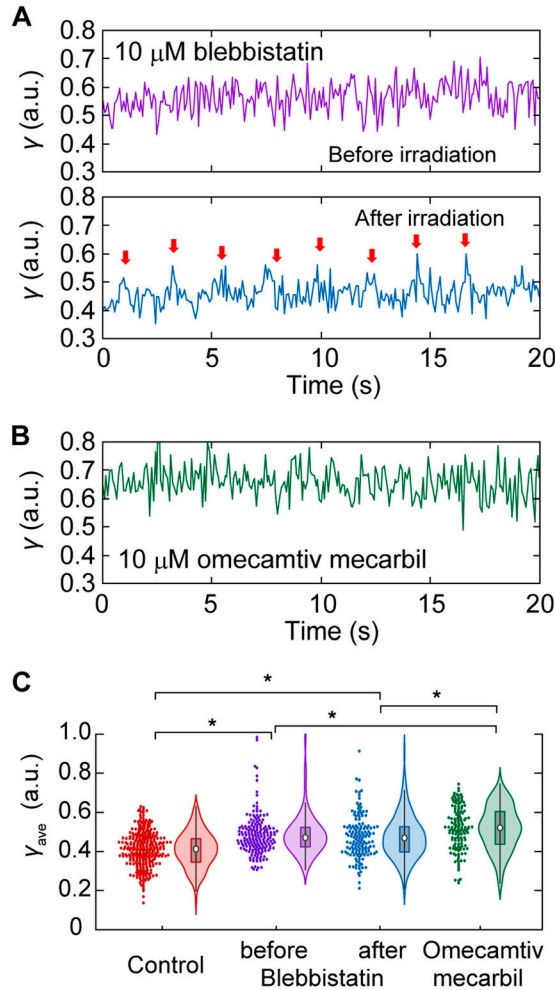

**Figure 4. Effect of drugs on γ-value during cardiomyocyte beating.**
**(A)** A typical trace of γ-value after 10 μm BS treatment (*upper*) and further after inactivation of BS efficacy by blue light irradiation (*lower*). Red arrows indicate each pulsation confirmed by eye inspection of the video. **(B)** A typical trace of γ-value after 10 μm OM treatment. **(C)** Violin plots of γ in the case of no chemical treatment (*red*, N = 295 traces), 10 μm BS treatment (*magenta*, N = 176), after inactivation of 10 μm BS treatment (*blue*, N = 143), and 10 μm OM treatment (*green*, N = 141). Asterisks indicate less than 0.01 of the *P*-value in the *t* test.

phenotype of genetic cardiomyopathy. Mutations in the *MYBPC3* gene encoding cardiac myosin-binding protein C (cMyBPC) are among the most common gene mutations found in hypertrophic cardiomyopathy (Flashman et al, 2004; Carrier et al, 2015). The cMyBPC function is an inhibitory control of crossbridge formation by bridging between myosin and actin. Here, we established a hiPSC line with a heterozygous frameshift mutation from a patient with hypertrophic cardiomyopathy caused by an *MYBPC3* mutation (*Mybpc3*[t/+]). For comparison, hiPSC lines with a homozygous mutation (*Mybpc3*[t/t]) as a positive control or without mutations (*Mybpc3*[+/+]) as a negative control were also established with a genome editing technique. We then measured the time course of SHG polarization emitted from sarcomeres in cardiomyocytes derived from these cell lines (Fig 5A).

Spontaneous pulsations were observed in cardiomyocytes differentiated from all established hiPSCs (Fig 5B). We obtained the $\gamma_{rest}$ and $\gamma_{cont}$ values in each 40-s time course and collected the γ-values from the FOVs, including a total of 60–120 sarcomeres from 17 cells. Both the mean $\gamma_{rest}$ (0.49 ± 0.06) and mean $\gamma_{cont}$ (0.57 ± 0.06) of *Mybpc3*[t/t] were significantly higher than those of *Mybpc3*[t/+] ($\gamma_{rest}$ = 0.40 ± 0.06, $P$ = 8.6 × $10^{-4}$; $\gamma_{cont}$ = 0.52 ± 0.07, $P$ = 7.6 × $10^{-3}$ in $t$ test) (Fig 5C, *red and blue*). *Mybpc3*[+/+] exhibited a lower $\gamma_{cont}$ (0.47 ± 0.05, $P$ = 5.7 × $10^{-3}$ in $t$ test) than *Mybpc3*[t/+], whereas there was no significant difference in $\gamma_{rest}$ between the two cells (0.42 ± 0.45, $P$ = 0.73 in $t$ test) (Fig 5C, *green*). The difference between $\gamma_{cont}$ and $\gamma_{rest}$, denoted as $d\gamma$ = $\gamma_{cont}$ − $\gamma_{rest}$ was thought to reflect the population of myosin actively interacting out of the inhibitory control. We found no significant difference between *Mybpc3*[t/+] and *Mybpc3*[t/t] in $d\gamma$ (Fig 5D, *red and blue*). Meanwhile, significant differences were observed between *Mybpc3*[+/+] and the other cell lines ($P$ = 3.9 × $10^{-4}$ in $t$ test) (Fig 5D, *green*). Between *Mybpc3*[t/t] and *Mybpc3*[+/+], there were significant differences in all $\gamma_{cont}$, $\gamma_{rest}$, and $d\gamma$ ($P$ = 4.8 × $10^{-6}$, 2.2 × $10^{-11}$, and 5.4 × $10^{-6}$, respectively, in $t$ test). Thus, the differences in γ measurement represent the effect of heterozygous and homozygous *MyBPC3* deficiency or the repair of actomyosin activity.

Although we expected that the contraction yield of sarcomeres would increase based on a previous experimental report that cellular shortening increased with *MyBPC3* deficiency (Toepfer et al, 2019), obvious difference was not observed in contraction yield among the three cell lines (0.08 ± 0.06 for *Mybpc3*[t/+], 0.08 ± 0.04 for *Mybpc3*[t/t], and 0.08 ± 0.05 for *Mybpc3*[+/+]) (Fig 5E, *lower*). The mean $SL_{rest}$ and $SL_{cont}$ were, respectively, 1.96 ± 0.11 μm and 1.79 ± 0.11 μm for *Mybpc3*[t/+], decreased after homozygous mutation ($SL_{rest}$ = 1.85 ± 0.16 μm, $P$ = 8.0 × $10^{-3}$; $SL_{cont}$ = 1.70 ± 0.15 μm, $P$ = 0.02 in $t$ test), and did not change after rescuing the phenotype ($SL_{rest}$ = 1.91 ± 0.12 μm and $SL_{cont}$ = 1.75 ± 0.12 μm) (Fig 5E, *upper*). Analysis of variance (ANOVA) for the three groups exhibited no statistically significant difference in $SL$ measurements ($P$ = 0.03 for $SL_{rest}$ and 0.08 for $SL_{cont}$). Although there might be a trend for homozygous mutations in shortening the $SL$, we were not able to show it statistically in this analysis.

Repairing this deficiency with genome editing decreased the slope of the γ–SL correlation (Fig 5F, *top and bottom*). Because a heterozygous *MyBPC3*-deficiency increases the probability of myosin–actin interaction during contraction (Toepfer et al, 2019), the slope of the γ–SL correlation can be attributed to the probability of crossbridge formation during contraction according to this result. In addition, the myosin in heterozygous *MyBPC3*-deficiency cardiomyocytes did not fully dissociate from actin in the resting state (Toepfer et al, 2019). Although the increases in $\gamma_{cont}$ and $\gamma_{rest}$ upon homozygous deficiency shown in Fig 5C were confirmed by the γ–SL correlation, there was no obvious difference in the slopes of the correlations between *Mybpc3*[t/+] and *Mybpc3*[t/t] (Fig 5F, *top and middle*). Relaxation of the γ-value was not affected by *MyBPC3* deficiency regardless of its homozygosity or heterozygosity (Fig 5G, *left*), indicating that cMyBPC is not directly responsible for the kinetics of the association–dissociation cycle of actomyosin corresponding to ATP hydrolysis. Meanwhile, the average trace of sarcomere movement exhibited a delay of relaxation in *Mybpc3*[t/+] and *Mybpc3*[t/t] in contrast to *Mybpc3*[+/+] cells (Fig 5G, *right*), which was consistent with previous results obtained in whole-cell shortening (Toepfer et al, 2019).

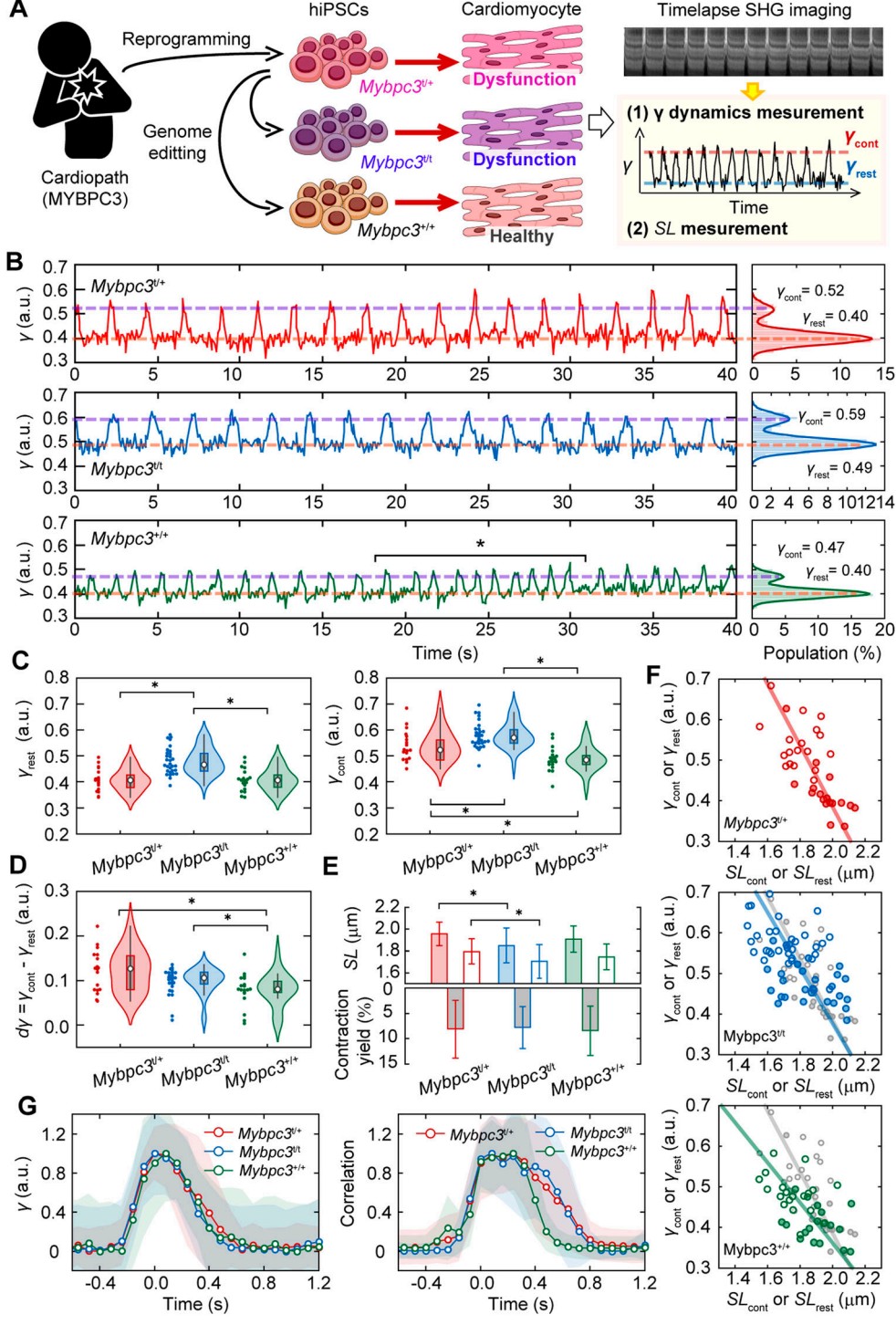

**Figure 5. SHG anisotropy measurement of cardiomyocytes differentiated from genetic disease patient-derived hiPSCs.**
**(A)** A schematic drawing of the experimental procedure. **(B)** A typical example of dynamic measurement of the sarcomere movement of parameter $\gamma$ with 80 ms time resolution in $Mybpc3^{t/+}$ (top), $Mybpc3^{t/t}$ (middle), and $Mybpc3^{+/+}$ cells (bottom). Right panels are histograms of $\gamma$-value in each trace. Solid lines in the right panels are fitting results with double Gaussian distributions. Magenta lines indicate $\gamma_{cont}$ and orange lines indicate $\gamma_{rest}$ in each trace obtained by the fitting. Asterisk indicates the occasional change of the beating cycle. **(C)** Violin plots of $\gamma_{rest}$ (left) and $\gamma_{cont}$ (right) for $Mybpc3^{t/+}$ (red, N = 21 FOVs), $Mybpc3^{t/t}$ (blue, N = 36), and $Mybpc3^{+/+}$ (green, N = 28). **(D)** Violin plots of $d\gamma_{ave} = \gamma_{cont} - \gamma_{rest}$ in $Mybpc3^{t/+}$ (red, N = 21), $Mybpc3^{t/t}$ (blue, N = 36), and $Mybpc3^{+/+}$ (green, N = 28). **(E)** Bar graphs of $SL$ in the resting state (filled) and in the contracting state (opened) (upper) and contraction yield (lower). Error bars are SD. **(F)** Relationship between $SL_{rest}$ or $SL_{cont}$ and $\gamma_{rest}$ or $\gamma_{cont}$ for $Mybpc3^{t/+}$ (top), $Mybpc3^{t/t}$ (middle) and $Mybpc3^{+/+}$ (bottom). Filled and opened marks are for the resting state and for the contracting state, respectively. Solid line is a line linked between averaged value of data for the resting state and for the contracting state. Overlapped gray marks and line in middle and bottom panels are those for $Mybpc3^{t/+}$. **(G)** Averaged time courses of $\gamma$-value (left) and sarcomere motion (right) and average trace of a pulsation in all analyzable data for $Mybpc3^{t/+}$ cell (red, N = 8), $Mybpc3^{t/t}$ cell (blue, N = 13), and $Mybpc3^{+/+}$ cell (green, N = 6). Light colors are SD. **(C, D, E)** Asterisks in (C, D, E) indicate less than 0.01 of the P-value in the t test.

## Detection of acquired myocardial dysfunction after photodamage

Next, we investigated the feasibility of SHG polarization anisotropy to detect acquired dysfunctions in cardiac myosin activity using an experimental model of radiation exposure. Even though ionizing radiation causes the apoptosis and necrosis of embryonic stem cells (ESCs), some ESCs partially survive and maintain their trilineage differentiation potential in humans and mice (Wilson et al, 2010; Hellweg et al, 2020). In mice, radiation-dosed ESCs could differentiate into cardiomyocytes; however, their beating was abnormal (Hellweg et al, 2020). We reproduced the acquired myocardial dysfunction using hiPSCs instead of mouse ESCs and ultraviolet (UV) irradiation instead of ionizing radiation and investigated the effect of UV irradiation on the $\gamma$-value and $\gamma$–$SL$ correlation (Fig 6A). HiPSCs

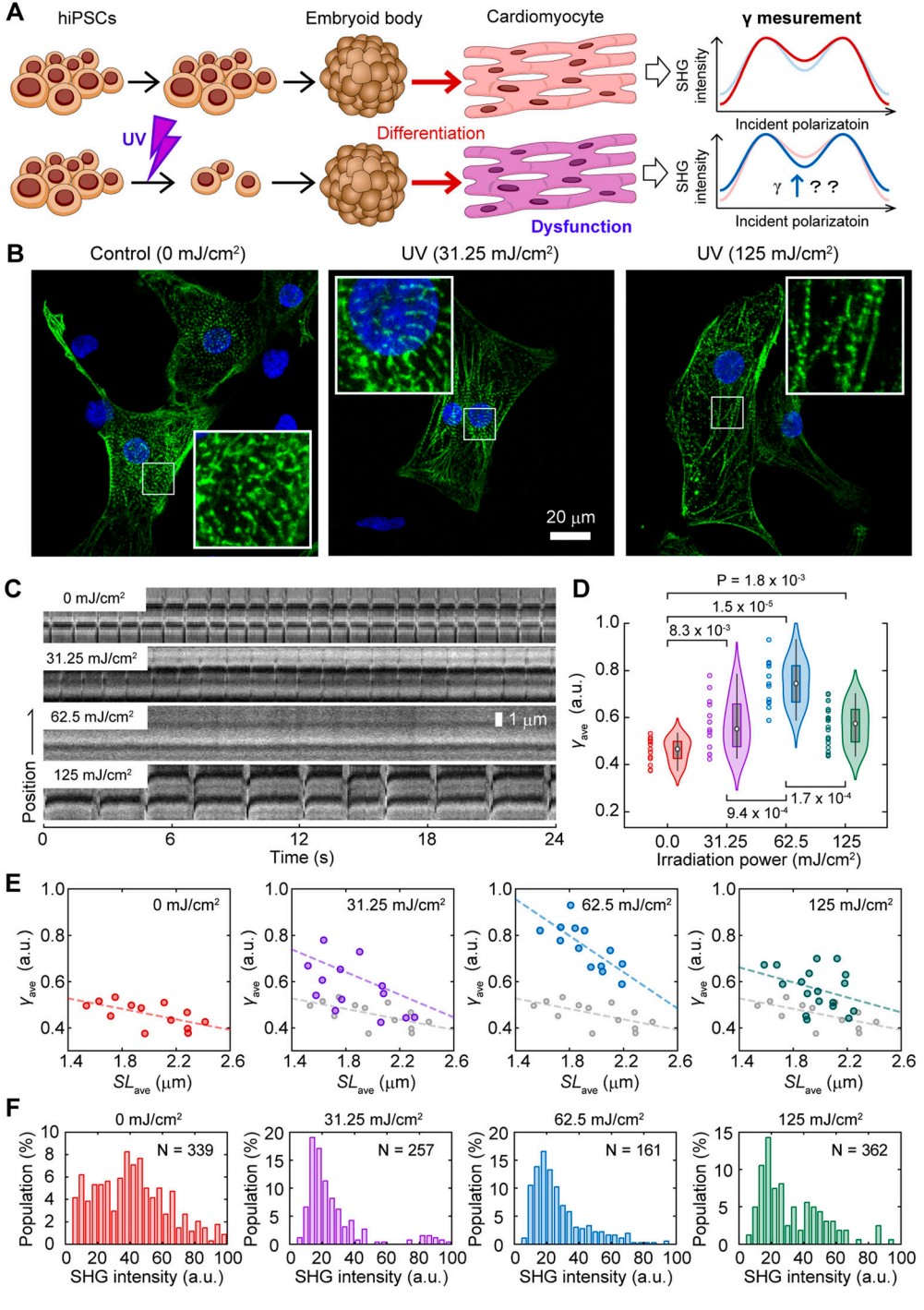

**Figure 6. SHG anisotropy measurement of acquired force generation dysfunction.**
**(A)** A schematic drawing of the experimental procedure. **(B)** Confocal fluorescent microscope image of cardiomyocytes stained with anti-α-actinin (*green*) and DAPI (*blue*). Cardiomyocytes derived from hiPSCs were irradiated by UV light (254 nm) at energies of 0.00 (*left*), 31.25 (*center*), and 125.00 mJ/cm² (*right*). The insets are enlarged images shown in a white rectangle in each image. **(C)** Kymographs of the SHG image of sarcomeres in cardiomyocytes derived from hiPSCs irradiated at 0.00 (*top*), 31.25 (*second top*), 62.5 (*second bottom*), and 125.00 mJ/cm² (*bottom*) by UV. **(D)** Violin plots of $\gamma_{ave}$ in a sarcomere in cardiomyocytes derived from hiPSCs irradiated at 0.00 (*red*, N = 13 FOVs), 31.25 (*magenta*, N = 13), 62.5 (*blue*, N = 13), and 125.00 mJ/cm² (*green*, N = 15) by UV. Asterisks indicate less than 0.01 of the P-value by the Mann–Whitney's U-test. **(E)** Correlation plots between $SL_{ave}$ and $\gamma_{ave}$ for 0.00 (*red*), 31.25 (*magenta*), 62.5 (*blue*), and 125.00 mJ/cm² (*green*). Solid line is a fitting result of data with a linear function. Overlapped gray marks and lines in the middle and bottom panels are those for 0.00 mJ/cm². **(F)** Histogram of average SHG intensity of a sarcomere in cardiomyocytes derived from hiPSCs irradiated at 0.00 (*red*), 31.25 (*magenta*), 62.5 (*blue*), and 125.00 mJ/cm² (*green*) by UV.

were irradiated with UV light (254 nm) at energy densities of 0.00, 31.25, 62.50, and 125.00 mJ/cm². Although cell viability after irradiation decreased with an increase in irradiation energy to 92%, 10.5%, 4.5%, and 0.5%, respectively, more than 90% of the surviving cells positively expressed the pluripotency markers Sox2 and Oct4 (Fig S4). These surviving irradiated iPSCs could differentiate via embryoid body (EB) formation into beating cardiomyocytes with or without UV irradiation.

Anti-α-actinin immunostaining showed the sarcomere structure in cells differentiated from UV-irradiated hiPSCs, indicating that these cells were capable of differentiating into cardiomyocytes. Some of the cells were negative for anti-α-actinin even in nonirradiated cells, indicating that our protocol for cardiomyocyte differentiation does not differentiate 100% of hiPSCs (Fig 6B). Even though the beating of the cardiomyocyte colonies seemed normal, the pulsations locally observed on the SHG microscope

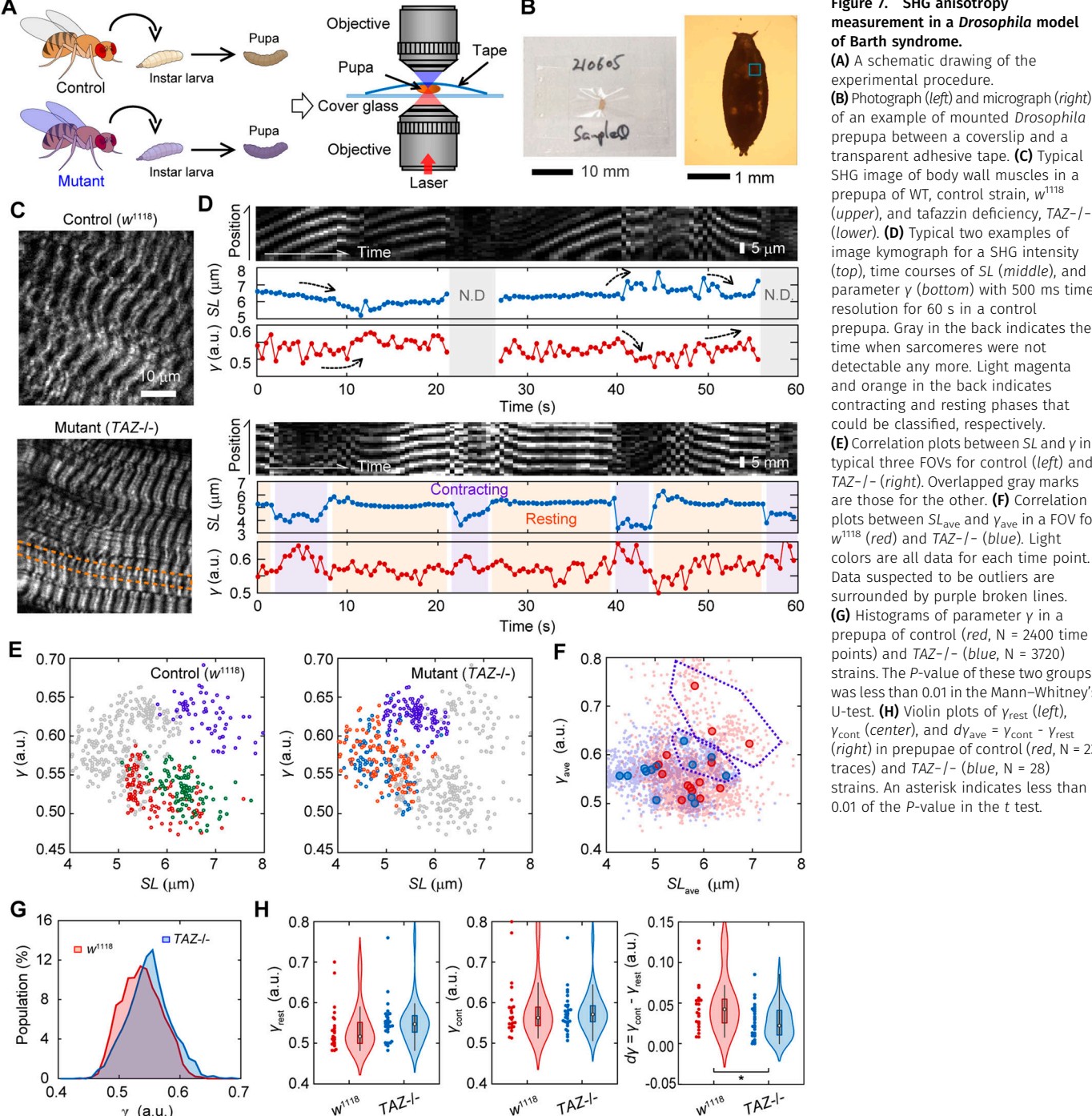

**Figure 7. SHG anisotropy measurement in a *Drosophila* model of Barth syndrome.**
**(A)** A schematic drawing of the experimental procedure. **(B)** Photograph (*left*) and micrograph (*right*) of an example of mounted *Drosophila* prepupa between a coverslip and a transparent adhesive tape. **(C)** Typical SHG image of body wall muscles in a prepupa of WT, control strain, $w^{1118}$ (*upper*), and tafazzin deficiency, *TAZ−/−* (*lower*). **(D)** Typical two examples of image kymograph for a SHG intensity (*top*), time courses of *SL* (*middle*), and parameter γ (*bottom*) with 500 ms time resolution for 60 s in a control prepupa. Gray in the back indicates the time when sarcomeres were not detectable any more. Light magenta and orange in the back indicates contracting and resting phases that could be classified, respectively. **(E)** Correlation plots between *SL* and γ in typical three FOVs for control (*left*) and *TAZ−/−* (*right*). Overlapped gray marks are those for the other. **(F)** Correlation plots between $SL_{ave}$ and $γ_{ave}$ in a FOV for $w^{1118}$ (*red*) and *TAZ−/−* (*blue*). Light colors are all data for each time point. Data suspected to be outliers are surrounded by purple broken lines. **(G)** Histograms of parameter γ in a prepupa of control (*red*, N = 2400 time points) and *TAZ−/−* (*blue*, N = 3720) strains. The *P*-value of these two groups was less than 0.01 in the Mann–Whitney's U-test. **(H)** Violin plots of $γ_{rest}$ (*left*), $γ_{cont}$ (*center*), and $dγ_{ave} = γ_{cont} - γ_{rest}$ (*right*) in prepupae of control (*red*, N = 23 traces) and *TAZ−/−* (*blue*, N = 28) strains. An asterisk indicates less than 0.01 of the *P*-value in the *t* test.

were weakened after irradiation at 31.25 mJ/cm², and they became almost undetectable after UV irradiation at 62.25 mJ/cm² (Fig 6C). Surprisingly, sarcomere pulsation was still observed upon further irradiation (Fig 6C, *bottom*). The $γ_{ave}$ calculated, including both pulsating and non-pulsating sarcomeres, were 0.46 ± 0.05, 0.57 ± 0.11, 0.75 ± 0.10, and 0.57 ± 0.09 upon UV irradiation at 0.00, 31.25, 62.50, and 125.00 mJ/cm², respectively (Fig 6D).

The sarcomere length was not shortened upon UV irradiation, although the $γ_{ave}$ value increased (Fig S5). The slope of the relationship between the average *SL* ($SL_{ave}$) and $γ_{ave}$ changed depending on the irradiation power (Fig 6E). UV irradiation at 31.25 mJ/cm² increased both the slope and the intercept of the liner $γ_{ave}$–$SL_{ave}$ correlation (Fig 6E, *second left*); the slope and the intercept were further increased at 62.5 mJ/cm² irradiation (Fig 6E, *second right*); at 125 mJ/cm² irradiation, the slope returned to that

for no irradiation, whereas maintaining the increase in intercept (Fig 6E, *right*). SHG intensity decreased upon UV irradiation (Fig 6F).

### Intravital evaluation of force generation dysfunction in a *Drosophila* model of Barth syndrome

To study the effects of genetic mutations on muscle contraction and heart failure, it is desirable to conduct experiments in small animal models, such as *Drosophila*, which have a large library of gene deletions. Because one of the strong advantages of SHG microscopy is applicability to deep tissue imaging because of its two-photon excitation optics, the present method may be applicable to *Drosophila* imaging. We challenged to evaluate muscle dysfunction in a *Drosophila* model of Barth syndrome caused by mutations in the *tafazzin* gene (*TAZ*) (Xu et al, 2006; Pu, 2022). *TAZ* deficiency in humans results in cardiomyopathy, neutropenia, myopathy, growth retardation, and 3-methylglutaconic aciduria through a deficiency of the phospholipid cardiolipin (1,3-bis(sn-3′-phosphatidyl)-sn-glycerol) in the inner mitochondrial membrane (Zegallai & Hatch, 2021). In this study, we prepared a white mutant strain of *D. melanogaster*, $w^{1118}$, as a control and a homozygous *TAZ*-deficiency mutant strain (*TAZ−/−*), which exhibits Barth syndrome-related phenotypes, such as having a significant reduction in cardiolipin, mitochondrial abnormalities, and the degradation of muscle activity (Xu et al, 2006). A third instar larva from each fly strain was grown to the prepupa stage and mounted on coverslips with transparent adhesive tape to measure the SHG polarization anisotropy (Fig 7A and B). *Drosophila* prepupae are considered technically challenging samples for optical observations because of their low transparency and the difficulty in controlling their spontaneous movement.

SHG microscopy visualized the sarcomeres of the body wall muscles in *Drosophila* prepupae without uncovering the puparium, regardless of the presence or absence of *TAZ* (Fig 7C). Unlike in the cardiac muscle, the contractions in *Drosophila* prepupae did not occur periodically. Instead, spontaneous muscle activity occurs randomly everywhere at all times in a prepupa. Hence, we extended the measurement duration to 60 s with the time interval to 500 ms, and expanded the FOV to 25 $\mu$m × 20 $\mu$m with the spatial resolution of 1 $\mu$m/pixel, to capture the random muscle contraction. Even in a prepupa, we could confirm an antiparallel manner of $\gamma$–$SL$ correlation in spontaneous contraction or relaxation rather than regular pulsations (Fig 7D, *upper*, *arrows*), except during the times when sarcomeres move out of the focal plane during observation (Fig 7D, *upper*, *gray*). Occasionally, clear state transitions from resting to contracting or from contracting to resting could be pursued (Fig 7D, *lower*). The $SL$ and $\gamma$-values within a single FOV were negatively correlated in both control and *TAZ−/−* flies (Fig 7E), as in the case of hiPSC-derived cardiomyocytes (Fig 3D). Plotting the data obtained from 14 FOVs for the control and 13 FOVs for *TAZ−/−* with the same coordinates, most of the data points seemed to be linearly correlated (Fig 7F). Meanwhile, minor populations with longer $SL$ showed a different correlation (Fig 7E and F, *magenta*).

The value of $SL$ (~7 $\mu$m) for the control prepupae (Fig 7E and F) was consistent with the previous reports (Prent et al, 2008) and was obviously different from that for *TAZ−/−* prepupae (Fig S6), which did not contradict that of a mice *TAZ*-deficiency model (Bertero et al, 2021). Meanwhile, we could not confirm the obvious difference in $\gamma$ time courses between them (Fig S7), but found only a tendency of *TAZ−/−* flies to be higher than the controls as shown in the histogram values of $\gamma$ (Fig 7G). A slight but significant difference was confirmed in the mean values of $\gamma$ (0.54 ± 0.03 for control; 0.55 ± 0.04 for *TAZ−/−*, $P = 0.013$ in a Mann–Whitney U-test). Then, we extracted the traces that included obvious contracting–resting transitions and estimated the $\gamma_{rest}$, $\gamma_{cont}$, and the difference between them, $d\gamma$. Although no statistically significant difference was observed, the mean $\gamma_{rest}$ for *TAZ−/−* flies was higher than that of the control, whereas the mean $\gamma_{cont}$ remained unchanged (Fig 7H, *left and center*). The mean $d\gamma$ was significantly lower in *TAZ−/−* group than in the control group (Fig 7H, *right*; $P = 0.01$ in a Mann–Whitney's U-test). Thus, the present method was applicable for investigating muscle activity in a *Drosophila* disease model.

## Discussion

In this study, we demonstrated the use of SHG anisotropy to investigate myocardial dysfunction in cardiomyopathy by constructing an effective assay system based on our highly sensitive SHG microscope with a fast polarization-controllable device (Kaneshiro et al, 2016; Kaneshiro et al, 2019), and the practical applications of this technique in examining dysfunctional cardiomyocytes differentiated from hiPSCs and a *Drosophila* disease model. Disease-derived and genome-edited cardiomyocytes provided information about the parameter $\gamma$ obtained from SHG anisotropy measurements, reflecting myosin force generation. The sarcomere length $SL$ was simultaneously measurable, and the correlation between $\gamma$ and $SL$ represented the phenotype of actomyosin dysfunction in genetically diseased cardiomyocytes. Furthermore, the present method was applicable for detecting acquired muscle failure because of radiation exposure and functional decline in *Drosophila* intravital imaging.

Two practical problems needed to be solved in measuring SHG anisotropy during myocardial beating. The first problem was the random orientation of myofibrils in cardiomyocytes, which limited the measurement of $\gamma$-values to well-aligned skeletal muscles. To remedy this, we employed a line-and-space pattern substrate, which linearly align the myofiber orientation in cardiomyocytes (Figs 1C and 2A), allowing SHG anisotropy analysis to be performed easily and reproducibly. The second problem is the low temporal resolution of current SHG polarization measurements. Cardiomyocytes beat within approximately 1 s. If the sarcomere orientation can be perfectly aligned at a given polarization angle, measurement at two or three polarization angles is sufficient to estimate the $\gamma$-value (Psilodimitrakopoulos et al, 2014; Forderer et al, 2016). However, unless perfectly adjusting the angle of myofiber orientation, which is challenging in cardiomyocytes, it would be practically difficult to estimate $\gamma$-values under the limited angle measurement. Our highly sensitive SHG microscope enabled the measurement of SHG anisotropy at 20 angles with a 10° angle pitch within 1 ms/pixel, which provided the SHG image acquisition with a temporal interval of 80 ms including 2–3 sarcomeres during myocardial beating. One additional issue has remained to carry

over to a future development that was controlling the pulsation cycle. Thereby, we occasionally observed fluctuations in beating cycles (Fig 5B, *bottom*, *asterisk*). Controlling the beating cycle needs the installation of an electrical stimulation system, buffer exchange system, and a temperature control system into a tiny space between two objectives with short working distance.

The values of parameters $\gamma_{relax}$ (0.43) and $\gamma_{rigor}$ (0.69) obtained in permeabilized cardiomyocytes (Fig 2B–D) were consistent with those obtained in other specimens by other groups (Plotnikov et al, 2006; Nucciotti et al, 2010; Schurmann et al, 2010). The $\gamma_{rest}$ (0.43) obtained in a living cell corresponded to the $\gamma_{relax}$ in permeabilized cells, and the $\gamma_{cont}$ (0.50) obtained was also a reasonable value, considering the active percentage (20–30%) of myosin population binding to actin during contraction (Fig 3C–G). However, these values were far different from those previously reported during isometric–tetanic contraction in single isolated fibers from frogs ($\gamma_{rest}$ = 0.30 and $\gamma_{cont}$ = 0.64) (Nucciotti et al, 2010). This discrepancy can be explained by the difference between tetanic contraction in skeletal muscle and cardiomyocyte pulsation: more myosins interact with actin during tetanic contraction, whereas only a small portion is activated in cardiac pulsation (Brunello et al, 2020; Hill et al, 2021). In collagenase-treated interosseous cells, it was reported that $d\gamma$ increased from 0.0 to 0.1 with a given electrical stimulus voltage (Forderer et al, 2016), which is in good agreement with the present results, $d\gamma$ = $\gamma_{cont}$ - $\gamma_{rest}$ = 0.07. Although all the $\gamma$-values obtained in this study did not contradict with the values shown in the previous literatures, we think that it is not worth comparing absolute $\gamma$-values in living cells between different experimental conditions. As discussed in an earlier study, absolute $\gamma$-values largely depend on the optics and calculation theories used (Schurmann et al, 2010), and also possibly depend on species of the observation sample. There should be a possibility that cell shape depends of $\gamma$-value, because $\gamma$ increases as the sarcomeres are forcibly shortened (Nucciotti et al, 2010) and that the sarcomere length depends on the surrounding environment, for example, cell shape (Bhana et al, 2010; Morris et al, 2020). It is more important to discuss common trends in the data obtained in the different experiments.

The attribution of $\gamma$, which has been controversial, is discussed here following the present results. SHG from muscles is dominated by SHG from myosin rather than actin. Plotnikov et al. argued that SHG from myosin is derived from the LMM and not from the S1 or S2 regions, according to their finding of small differences in SHG polarization in scallop muscles between upon AMPPNP treatment and that in the rigor state. They also did not observe any difference in SHG polarization from *C. elegans* muscles against the scallop muscle, whose ratio of paramyosin (headless myosin) to myosin is 15-folds higher in *C. elegans* muscles (Plotnikov et al, 2006). Meanwhile, Nucciotti et al. and Schürmann et al. independently claimed that the main contributor to SHG is the crossbridge state, according to their observations of the differences in SHG polarization dependence between conditions where myosin attaches to actin (the rigor state) and in conditions where myosin detaches from actin (the relaxed state) (Nucciotti et al, 2010; Schurmann et al, 2010). In this study, we also confirmed the difference in $\gamma$-values between the rigor and relaxed states in permeabilized cardiomyocytes and between the contracting and resting states

in pulsating cardiomyocytes. There is almost no doubt that SHG polarization corresponds to the crossbridge condition. It is additionally possible that the dipole orientation environment around the LMM linked to the actomyosin crossbridge condition contributes to the change in $\gamma$. The theoretical curve obtained using Equation (2) almost overlapped with the averages of the data obtained in the rigor state, ATP treatment, and AMPPNP treatment in the correlation plot between total intensity and $\gamma$ (Fig 1H, *gray broken line*). It is unlikely that the stable helical structure of LMM undergoes such a large change. This means that the primary cause of the changes in SHG polarization dependence is simply the angular change in the dipole orientation. Conclusively, it is reasonable to assume that S1 and S2 are the primary sources of change in SHG anisotropy.

Either way, the present results of the negative monotonous correlation of $\gamma$ and *SL* (Fig 3G), as previously reported in frog muscle fibers by Nucciotti et al. (Nucciotti et al, 2010), strongly support the previously proposed hypothesis that the $\gamma$-value indicates the ratio between attached to detached myosin heads (Nucciotti et al, 2010; Schurmann et al, 2010), which is a crossbridge population but not necessarily relating to force-generating capability. One difference was that the $\gamma_{cont}$ and the $\gamma_{rest}$ depended on *SL* in cardiomyocytes, (Fig 3G) whereas only $\gamma_{cont}$ did so during isometric force generation in frog muscle fibers (Nucciotti et al, 2010). Because the $\gamma$ obtained from myosin in the relaxed state in cardiac muscle was altered by external forces (Yuan et al, 2019), the myosin that remained undetached from actin even in the relaxed state during cardiac beating was also thought to contribute to the $\gamma$-value.

In the drug treatment experiments, both BS and OM treatments, which are promising candidates as treatments for cardiomyopathy (Bond et al, 2013; Teerlink et al, 2016; Roman et al, 2018), increased $\gamma$-values, whereas stopping myocardial beating (Fig 4A and B). Because OM treatment slows down the lever arm swinging of myosin, resulting in myosin being unable to dissociate from actin (Planelles-Herrero et al, 2017; Woody et al, 2018), the present result for OM treatment can be explained by the above hypothesis. The increase in $\gamma$ upon OM treatment approached a value close to $\gamma_{rigor}$ (Fig 4C, *green*), which agrees with the previous result wherein OM increased $Ca^{2+}$ sensitivity in cardiomyocytes, causing more myosin to interact with actin (Swenson et al, 2017). Meanwhile, BS treatment inhibited product release during the ATP hydrolysis cycle, resulting in myosin dissociation from actin without any interference from structural changes in the lever arm (Kovacs et al, 2004). According to the above hypothesis, BS treatment causes a $\gamma$ decrease coupled with myosin dissociation. The actual results were contrary to this expectation and required a different or additional interpretation. It has been reported that the S1–S2 region of myosin bound to BS was more parallel to the filament axis of a myosin filament than without BS (Kampourakis et al, 2018), decreasing the dipole polarity angle $\varphi$ in Equation (1). In short, BS treatment is expected to cause an increase in $\gamma$ even in nonattached myosin lying along the filament axis. A similar result was previously reported, wherein N-benzyl-p-toluene sulfonamide (BTS), another myosin inhibitor that suppresses force generation without interfering with myosin binding to actin, caused a decrease in $\gamma$ (Schurmann et al, 2010). The $\gamma$-value only reflects the angle of S1 or S2 to the filament axis and is not

always determined by the ratio of the populations of attached and detached myosin in specific cases, such as in chemical drug treatments. Moreover, the increase in $\gamma$ upon BS treatment did not reach $\gamma_{rigor}$, and the inactivation of the effect of BS treatment upon blue light irradiation restored the $\gamma$-value and the beating behavior (Fig 4A and C), indicating that BS only affected myosin under physiological conditions. This result does not contradict a previous report that the inhibitory effect of BS in the myocardium was not associated with calcium influx but only acted on myosin (Dou et al, 2007). In situations wherein the mechanism of a drug's effect on actomyosin is clarified, $\gamma$ measurement is expected to be a powerful tool for evaluating drug efficacy.

The first demonstrative application was one of genetic cardio-myopathy, *MyBPC3* deficiency. An additional deficiency from the heterozygous mutant to homozygous caused an increase in $\gamma_{cont}$ and $\gamma_{rest}$ without changing the slope of the $\gamma$–SL correlation (Fig 5C and F, *red and blue*). It can be interpreted that the $\gamma$-value in *Mybpc3*$^{t/t}$ was positively offset from that in *Mybpc3*$^{t/+}$ because of the constant residual myosin interacting with actin. Rescue experiments caused a decrease in the slope of the $\gamma$-SL correlation without increasing the $\gamma_{rest}$ (Fig 5C and F, *green*). There are two possible causes for the increase in $\gamma_{rest}$ according to the interpretation of the present result shown in Fig 5; in the resting state, myosin heads lying on the filament axis or the forcible interaction of myosin with actin. As previously reported, a deficiency in cMyBPC induces excessive myosin–actin interaction, causing an increase in the contractile force generated and insufficient relaxation: a heterozygous deficiency enhances crossbridge formation, whereas a homozygous deficiency further inhibits super relaxation, which is the state with a 10-folds slower ATPase cycle than myosin in the conventional relaxed state (Toepfer et al, 2019). Considering the mechanisms underlying the effects of *MyBPC3* deficiency that the heterozygous mutation inhibited the super relaxation of myosin in relaxed state (Korte et al, 2003; Toepfer et al, 2019), the latter is more plausible. The increase or decrease in the slope of the $\gamma$–SL correlation can be interpreted to reflect the probability of unbound myosin binding to actin during contraction. Because the slope of the $\gamma$–SL correlation corresponds well to the generated force, as previously reported (Nucciotti et al, 2010), this result is consistent with the finding that a *MyBPC3* deficiency enhances the contraction force (Toepfer et al, 2019). Interestingly, *MyBPC3* deficiency slowed myosin relaxation without changing $\gamma$ dynamics (Fig 5G), indicating a decoupling of myosin dissociation and muscle relaxation. In summary, the $\gamma$-related values (Fig 5C and D), $\gamma$–SL correlation (Fig 5F), and the average trace of $\gamma$ and sarcomere dynamics (Fig 5G) represented the phenotype of muscle dysfunction in *MyBPC3*-deficiency cardiomyocytes, suggesting that the two parameters SL and $\gamma$ could be used as parallel indicators to determine the mechanism of actomyosin dysfunction. Especially, the probability and frequency of crossbridge formation for force generation could be expressed by the slope of $\gamma$–SL correlation. It should be noted that the $\gamma$-value is only measured locally, and the sarcomere motion reflects the entire myofibril behavior in this aspect. The prolonged relaxation time mediated by the *MyBPC3* deficiency would not be because of myosin dissociation but rather decreased myosin cooperativity or synchrony. Although a more detailed verification is needed in the future to assess the

reliability of this hypothesis, the results we obtained demonstrate that the present method allows the simultaneous evaluation of myosin dynamics both on the macro- and micro-scales. We therefore concluded that the present method is applicable for evaluating actomyosin dysfunction caused by *MyBPC3* deficiency.

Subsequently, we attempted to detect acquired myocardial dysfunction using cardiomyocyte differentiated from UV-irradiated hiPSC (Fig 6A–C). In this application, we estimated the $\gamma_{ave}$ as an indicator to compare actomyosin activity between samples containing beating and non-beating cardiomyocytes. Curiously, $\gamma_{ave}$ increased depending on the irradiation power up to 62.5 mJ/cm$^2$ and decreased irradiation by 125 mJ/cm$^2$ (Fig 6D). This can be explained by assuming the presence of two cell populations: cells with strong or acquired resistance to irradiation and cells damaged by irradiation. In addition, it was also assumed that the latter population is far smaller than that of the former. At 31.25 mJ/cm$^2$ irradiation, the data were mainly obtained in the damaged cell population. The damaged cells received even more damage when increasing the irradiation power to 62.5 mJ/cm$^2$. A stronger irradiation dose, 125 mJ/cm$^2$, killed the damaged cells, so irradiation-resistant cells were mainly detected. This interpretation is consistent with the results of the intensity histograms (Fig 6F): the population showing high SHG intensity that disappeared at 31.25 mJ/cm$^2$ irradiation reappeared with further increases in irradiation intensity. In either case that UV irradiation increased the myosin population lying along the filament axis or continuing to bind to actin without force generation, the base of the $\gamma$ change could be attributed to the inactivated or dead myosin. Again, the increase in the slope of the $\gamma_{ave}$–SL correlation is because of the promotion of crossbridge formation during muscle contraction. The result that both the slope and intersect of the $\gamma_{ave}$–SL correlation were increased (Fig 6E), UV irradiation might cause not only the direct inactivation of myosin function but also the indirect effect on its force generation similar to *MyBPC3* deficiency. According to the monotonic correlation between the total SHG intensity and $\gamma$ shown in Equation (2), the SHG intensity increases as $\gamma$ increases with the irradiation power. Contrary to this expectation, the intensity of SHG emitted from the sarcomeres decreased upon UV irradiation (Fig 6F). It can be speculated that UV irradiation decreases the number of myofibrils because of the malfunction of myofibril bundling. This trial experiment demonstrated that the $\gamma$-value measurement allowed the evaluation of myocardial dysfunction even in cases in which sarcomere contraction rarely occurs. This emphasizes the importance of using $\gamma$-values rather than the SHG intensity to evaluate actomyosin activity. The mechanism of the acquisition of dysfunction is not the focus of this study, but it is undoubtedly an interesting research prospect in radiation biology.

Finally, we applied the present method to a challenging sample: intravital imaging using a *Drosophila* disease model of *TAZ* deficiency. Although we observed a difference in $d\gamma$ in muscle failure because of *TAZ* deficiency, some technical problems remained. First, it was difficult to select an observation area. It has been reported that regions where the sarcomeres intersect, such as those shown in Fig 7C, *orange broken lines*, are illusory optical artifacts (Dempsey et al, 2015). With sarcomeres constantly moving in all directions, these areas could not be avoided. Furthermore, depending on the stretching conditions and other factors, the

sarcomeres may appear double peaked or single peaked in SHG observation (Prent et al, 2008). Thereby, the accuracy of the SHG measurement should be lower in this experiment than that in experiments using hiPSC-derived cardiomyocytes. It was also difficult to quantify the *SL* in a living *Drosophila* sample, because the sarcomeres in the FOV did not contract uniformly at the same time, and the illusory optical artifacts mentioned above also may cause false contrasts. Adopting other methods, such as using diffraction, is required to fulfill this aim. Nevertheless, we could obtain the important information to discuss the mechanism of myocardial dysfunction induced by *TAZ* deficiency. According to the previous literatures, the *TAZ*-deficiency disorders mechanochemical coupling of myosin force generation via abnormal $Ca^{2+}$ circulation induced by accumulation of reactive oxygen product (Liu et al, 2021). Meanwhile, because cardiolipin promotes ATP synthesis by stabilizing the protein complexes for the electron transport circuit on the mitochondrial membrane, the *TAZ* deficiency degrades the efficiency of ATP production by decreasing ATP synthase activity in hiPSC models (Wang et al, 2014) or by suppressing the formation of dimmer rows of ATP synthase in the *Drosophila* model (Acehan et al, 2011). The resulting ATP depletion would reduce myosin activity. However, Wang et al. concluded that hiPSC-derived cardiomyocyte with *TAZ* deficiency contains a critical defect in contractile force independent of ATP depletion based on their own result that inhibition of mitochondrial ATP production in healthy cardiomyocytes has no effect on contractility (Wang et al, 2014). Summarizing the present results of a *Drosophila* Barth syndrome model, *TAZ* deficiency induced the shortening of sarcomeres and the decrease of *dγ*, most likely because of the increase of $γ_{rest}$, but did not impact the *γ–SL* correlation (Fig 7E). Variation of active myosin but not inactive myosin population should be expressed as a change in the slope of *γ–SL* correlation according to the previous reports (Plotnikov et al, 2006; Nucciotti et al, 2010; Schurmann et al, 2010) and the present results in Figs 5 and 6. Therefore, we can claim that *TAZ* deficiency promotes the increase of inactive myosin population binding to actin, like in the rigor state, but not active myosin. If ATP depletion is not involved, as Wang et al. claimed, inhibitory regulation to myosin would be promoted via disturbed $Ca^{2+}$ flux in *TAZ*-deficiency mutant. The present results in the *Drosophila* model demonstrates the applicability of the present method for deep tissue imaging as a major step forward in cardiomyopathy research.

Finally, we discuss the extensibility of SHG anisotropy measurements in actomyosin studies. Yuan et al. recently found that SHG anisotropy is attributed not only to the electric dipole along the filament axis but also to the reduction in crystallographic symmetry (Yuan et al, 2019). Crystallographic symmetry is represented as an asymmetric feature of SHG polarization. The formula for its approximate calculation was updated as follows:

$$I(\theta) = A\left[\left\{\gamma\cos^2(\theta-\theta_0) + \sin^2(\theta-\theta_0)\right\}^2 + \left\{\sin 2(\theta-\theta_0) + \delta\sin^2(\theta-\theta_0)\right\}^2\right]$$
(3)

where *δ* is a parameter reflecting the crystallographic symmetry. They observed a correlation between the *δ*-value and the ratio of two myosin isoforms, *α* and *β* (Yuan et al, 2019). Asymmetric features were also observed in this study (Fig S8A). Human atrial cardiomyocytes express two myosin isoforms, *α* and *β*, whereas ventricular cardiomyocytes express only *β*-myosin (Miyata et al, 2000). In the current cardiomyocyte differentiation protocols, pacemakers, atria, and ventricles are usually generated from the same stem cells in vitro (Yechikov et al, 2016). Because the present study did not perform subtype segregation, we might have unintentionally evaluated atrial cardiomyocytes that exhibited asymmetric features. Consequently, the distribution of the obtained *δ*-values was slightly positively biased (Fig S8B) and there were significant differences in *δ* among the rigor, relaxed, and AMPPNP-treated states (Fig S8C). Because the strong interaction of myosin with actin filaments loads rotational tension onto myosin filaments, altering the crystallographic symmetry (Huxley et al, 1983), the decrease in *δ* upon the addition of ATP or AMPPNP was speculated to be because of the release of rotational tension resulting from the dissociation of myosin from actin. Although the attribution of *δ*-values remains controversial, there is no doubt that *δ* measurements, along with *γ* measurements, are useful for quantifying the muscle condition. The use of *δ* measurements may improve the precision of the subtype identification. This advantage is expanding the application of the present method in the diagnosis of cancers, fibrosis, diseases involving the cornea, and tissue engineering (James & Campagnola, 2021).

In conclusion, this study demonstrated the applicability of SHG anisotropy to evaluate the phenotypes of genetic or acquired cardiomyopathy and quantitatively investigate its impact on actomyosin activity in cell models and a *Drosophila* model using the SHG anisotropic parameters reflecting the population of actively working crossbridges. The present method cannot directly measure the actual force exerted by the muscle but only compare the averaged *γ* and/or *γ–SL* correlation. However, quantitative comparison is valuable when evaluating the effect of drugs or genetic defects. The proposed method is applicable to various samples. In particular, it is expected to be useful and effective in iPSC research. The cell-based disease models based on hiPSC technology promise to accelerate the understanding of pathogenic mechanisms, the development of regenerative medicine, drug toxicity screening, and drug discovery (Grskovic et al, 2011; Bellin et al, 2012). To evaluate drug toxicity, disease severity or the efficacy of a drug against hiPSC-derived cardiomyocytes, measurements of myocardial activity, for example, muscle force generation, are required (Miyagawa & Sawa, 2018; Sewanan & Campbell, 2020). Considering the general use of hiPSCs, it is desirable to use quality-checked cells directly for this purpose. SHG microscopy, which is a nonstaining and noninvasive method, has an advantage in this regard. In fact, a technique has been proposed to identify subtypes of hiPSC-derived cardiomyocytes by noninvasively measuring the total SHG intensity of each cell, even without staining (Chang et al, 2020). Although construction and adjustment of microscope setup needs specialization in microscopy, once the microscope is built, the operation itself is simple and reproducible which will be ideal for general use. We hope that the present method will be an essential tool in mechanobiology, cardiomyopathy, and iPS research in the future.

## Materials and Methods

### iPSC culture and cardiomyocyte differentiation

The 253G1 human iPSC line was purchased from the Riken Cell Bank (HPS0002) and was adapted to feeder-free conditions as described previously (Nakagawa et al, 2008). The $Mybpc3^{t/+}$ hiPSC strain was established from a patient, and the $Mybpc3^{t/t}$ and $Mybpc3^{+/+}$ strains were edited using CRISPR-Cas9, as previously reported (Takeda et al, 2020). The cells were maintained on iMatrix511-silk-coated (892021; Matrixome) for 253G1 or iMatrix-511-coated (892012; Matrixome) dishes for $Mybpc3^{t/+}$, $Mybpc3^{t/t}$, and $Mybpc3^{+/+}$ strains in StemFit AK02N medium (Ajinomoto) at 37°C and 5% $CO_2$. The medium was exchanged every day, and passaging was performed twice per week.

For cardiomyocyte differentiation, hiPSCs were harvested with TrypLE select (12563011; Gibco), transferred to ultra-low attachment U-bottom 96 well plate (MS-9096U; Sumitomo Bakelite) at a density of $1 \times 10^4$ cells/well, and then acclimated to the mTeSR1 culture medium (ST-85850; STEMCELL Technologies) until differentiation began. Differentiation was induced 4 d after EB formation using the PSC Cardiomyocyte Differentiation Kit (A2921201; Gibco), following the differentiation time course described in the manufacturer's instructions. 2 d after the induction, EBs were transferred from the 96 well plate to a gelatin/iMatrix/fibronectin triple-coated dish (concentrations: 0.1%, 0.5 $\mu g/cm^2$, and 1 $\mu g/cm^2$, respectively) and allowed to differentiate up to 14 d after the induction.

### UV irradiation

HiPSCs on culture dishes were irradiated with 254 nm ultraviolet light (UV) for 0, 2.5, 5.0 or 10 min. The UV dose was measured with an S425C thermal power sensor equipped on a PM100D monitor (Thorlabs Japan) and was calculated to be 12.5 $mJ/cm^2/min$. Under these conditions, ~92%, 10.5%, 4.5%, and 0.5% cells survived, respectively (n = 2). These cells were cultured for more than two weeks after UV irradiation and used for subsequent experiments. We confirmed that more than 90% cells remained to express both pluripotency markers Sox2 and Oct4 after 10 passages with immunostaining analysis (93, 97, 95, and 99% for Sox2, and 99, 99, 100, and 99% for Oct4, in 300, 67, 351, and 100 cells, respectively) (Fig S4).

### Immunocytochemistry

Immunostaining was performed 14 days after the induction of differentiation. Cultured cells were washed twice with PBS(–), dissociated by trypsinization (0.25% Trypsin/EDTA, 25200056; Thermo Fisher Scientific), and replated on iMatrix511-coated cover glass (C218181; Matsunami) in a culture medium consisting of DMEM (FujiFilm Wako Pure Chemicals) with 10% FBS (DMEM/10% FBS). 4 d after the plating, cells were fixed with 4% PFA for 15 min at RT. After washing, the cells were permeabilized with 0.3% Triton X-100 in PBS(–) and then incubated with anti-$\alpha$-actinin mouse monoclonal antibody (1:100, A7732; Sigma-Aldrich) in CAS-Block reagent (008120; Life Technologies) at 4°C overnight. After washing, immunoreactive cells were determined using the appropriate fluorescently labeled secondary antibodies, Alexa Fluor 488 conjugated goat anti-mouse IgG (1:500, A11001; Molecular probes). The cells were washed, stained with DAPI (1 $\mu g$/ml, D9542; Sigma-Aldrich), and mounted in Vectashield mounting medium (H-1000; Vector Laboratories).

### Seeding cardiomyocytes on a micropatterned substrate

A strong dependence of myofibril orientation in cardiomyocytes on cell shape has been previously reported (Bhana et al, 2010; Morris et al, 2020). Cell shape can be easily controlled by culturing cells on a substrate on which adhesion areas are biochemically controlled (Bray et al, 2008). To make approximately one-dimensional myofibril arrays along the line direction, we cultured all cardiomyocytes on a line-and-space patterned glass substrate with a line width of 20 $\mu m$ and a space between adjacent lines of 300 $\mu m$ (CytoGraph L20S300; Dai Nippon Printing Co. Ltd.) (Fig 1C).

For seeding of 253G1-derived cardiomyocytes, the line-and-space patterned glass substrates were placed in six well cell culture plate (3516; Corning) with the cell adherent side up, coated with iMatrix 511-silk (0.5 $\mu g/cm^2$) in PBS(–) at 4°C overnight, and then washed three times with PBS(–). For $Mybpc3^{t/+}$, $Mybpc3^{t/t}$, and $Mybpc3^{+/+}$-derived cardiomyocytes, the patterned glass substrates were coated with 0.1% gelatin instead of iMatrix 511-silk in PBS(–) at 37°C for more than 30 min, and the excess gelatin solution was aspirated before seeding. On 14 d after the differentiation induction, the cardiomyocytes differentiated from hiPSCs were dissociated with 0.25% trypsin/EDTA for 10 min and resuspended in DMEM/10% FBS. Large cell aggregates were removed using a cell strainer (100 $\mu m$, 352360; Corning). The remaining cells were seeded on an iMatrix-coated or gelatin-coated line-and-space patterned glass substrate at a density of $4 \times 10^4$ cells/$cm^2$. The culture medium was exchanged every other day until 18 d after the induction, when the SHG analysis was performed.

### Preparation of permeabilized cardiomyocytes

To investigate the nucleotide dependence of the actomyosin structure, permeabilized cardiomyocytes were prepared following a previously reported protocol (Sato et al, 2017). Briefly, the surface membrane of cardiomyocytes cultured on a patterned substrate was removed with an extraction buffer (30 mM imidazole, pH 7.5, 70 mM KCl, 1 mM EGTA, 2 mM $MgCl_2$, 0.5% Triton X-100, and 4% polyethylene glycol [mol wt 8,000]) for 4 min on ice, then washed twice with PBS(–). The permeabilized cells were then treated on ice with the reported solutions to induce rigor and relaxation (5 mM ATP) states (Nucciotti et al, 2010). The AMPPNP state was induced using the same solution as that used for rigor with 5 mM AMPPNP.

### Drug treatment of cardiomyocytes

(S)-(–)-Blebbistatin (BS) was purchased from Toronto Research Chemicals Inc. (B592500, lot no. 5-FRU-48-2), dissolved in dimethyl sulfoxide to prepare a 10 mM stock solution, and stored at –80°C. Omecamtiv mecarbil (OM) was purchased from Toronto Research Chemicals Inc. (C544000, lot no. 1-TIM-105-1), dissolved in DMSO to make a 10 mM stock solution, and stored at –80°C. Cardiomyocytes

differentiated from 253G1 hiPSCs through EB formation were plated on a patterned substrate coated with iMatrix, as mentioned above, and were incubated at 37°C in the DMEM/10% FBS with 10 $\mu$M of BS or 10 $\mu$M of OM with 0.1% DMSO for 60 min. As the control sample against drug-treated cells, cardiomyocytes differentiated from 253G1 hiPSCs were incubated at 37°C with 0.1% DMSO for 60 min. All biological assays were repeated at least thrice.

### Preparation of *Drosophila* prepupa sample

The *TAZ−/−* strain was established following the previous report (Xu et al, 2006). A chromosome with null *TAZ* mutation was balanced over the CyO (Curly of Oster) chromosome tagged with GFP. The pupae were raised at 25°C until the third instar larval stage. GFP-negative larvae were counter-selected as homozygous flies for *TAZ*. SHG images were acquired during the prepupal stage at a controlled room temperature of 23°C. A prepupa was placed on a glass slip such that its dorsal surface was in contact with the glass surface. The sample was then immobilized from the ventral surface using transparent adhesive tape (Fig 7B).

### Optical setup of SHG microscope

We used the same optical setup of the SHG microscope as that used in previous studies (Kaneshiro et al, 2016; Kaneshiro et al, 2019). The brief descriptions are provided below. The laser source was a Ti: sapphire pulsed laser with a wavelength of 810 nm, pulse width of 200 fs, and a repetition rate of 80 MHz. The polarization state of the incident beam was controlled using a polarization controller consisting of a pair of electro-optic modulators. The incident beam was focused by an air-immersion objective lens with a magnification of 40× and a numerical aperture (NA) of 0.95. The incident beam was raster scanned in the observation plane by *x-y* galvanometer mirrors. Forward-scattered photons were detected by a photon-counting photomultiplier tube after collection by an objective lens (with a magnification of 60× and NA of 1.42 for cardiomyocytes, and with a magnification of 60× and NA of 0.70 for *Drosophila* prepupae), passed through a tube lens and a band-pass filter. Optical parts composing the microscope, including silver-coated mirrors, dielectric-multilayered dichroic mirrors, oxide-coated half-mirrors, anti-refection-coated lens, and so on, cause polarization distortion on the sample plane. We conducted the calibration of incident polarization dependence on detected intensity with an estimation of the phase retardation parameters without a sample (Kaneshiro et al, 2016).

### Image data acquisition

A set of polarization-resolved SHG images was acquired by synchronous control of the *x-y* scanner and the polarization controller. For all the measurements, the average power of the incident laser was set to 26 mW. This value led to an approximate average illumination intensity of 3 MW/cm$^2$ at the focus. For cardiomyocytes, the laser focus was raster scanned in a rectangle of 25 × 50 $\mu$m$^2$ with 128 × 256 pixels, with a dwell time per pixel of 1 ms. The incident polarization was swept within the dwell time at every pixel position. The angle pitch was 10° and the number of angles was 20, which

covered a total angle range of over 180°. The total acquisition time per image was about 33 s. In the dynamic measurements on 253G1 and *MyBPC3* deficient cardiomyocytes, 500 polarization-resolved image sets within a rectangular region of interest of 0.39 × 7.81 $\mu$m$^2$ (2 × 40 pixels) were repeatedly acquired with a pixel dwell time of 1 ms and no interval time between images. In the dynamic measurement of UV-treated cardiomyocytes, the rectangular region was set to 0.195 × 4.88 $\mu$m$^2$ (2 × 50 pixels). The total acquisition time per 500 frames was 40 s. For the dynamic measurement in a *Drosophila* prepupa, the rectangular region was set to 25 × 20 $\mu$m$^2$ with the spatial resolution of 1 $\mu$m/pixel and the temporal resolution of 1 ms. The polarization sweeping condition was the same as that used for cardiomyocytes. A total of 120 polarization-resolved image sets were acquired within 60 s.

### Estimation of whole sarcomere motion with correlation calculation

In some experiments, *SL* measurement was not possible because of the deterioration of the SHG image during contraction. To estimate sarcomere motion in such cases, we used the correlation between images because SHG images in the resting state are similar, whereas those during contraction differ from resting SHG images. The correlation was calculated by first calculating and averaging Pearson's correlation between the given time t and the others. Then, resting time was identified using Otsu's binarization method and Pearson's correlation was calculated and averaged for images at a given time t and resting time images to give a correlation value.

### Ethics statement

Research plan related to the hiPSCs used in this study was approved by the Ethics Committee of RIKEN Center for Biosystems Dynamics Research (approval number: RIKEN-K1-2021-003). The use of patient-derived samples and genomic analysis were approved by the Ethics Committee of Osaka University Hospital, and written informed consent was obtained from all patients. This study conforms to the ethical guidelines for medical and health research involving human participants in Japan and all principles outlined by the Declaration of Helsinki.

## Supplementary Information

## Acknowledgements

We thank Kohei Kawaguchi (RIKEN) for experimental preparations and research members at Dai Nippon Printing Co., Ltd. for providing the line-and-space pattern substrates and advice on how to use them. The *TAZ* mutant strain was kindly gifted by Dr. Mindong Ren (Xu et al, 2006). This work was mainly supported by the Japan Agency for Medical Research and Development (grant number: 22bm0804008h0006 to TM Watanabe and S Miyagawa, and 22zf0127001h002 to E Kuranaga) and the Japan Science and Technology Agency (CREST: JPMJCR1852 to TM Watanabe and E Kuranaga),

and was partially supported by Research Foundation for Opto-Science and technology and Japan Society for the Promotion of Science (JSPS), Grant-in-Aid for Scientific Research (B) (grant number JP21H03599 to TM Watanabe), and Ministry of Education, Culture, Sports, Science and Technology (MEXT) KAKENHI (grant number JP21H05255 to E Kuranaga). We would like to thank Editage (www.editage.com) for the English language editing.

## Author Contributions

H Fujita: investigation, and writing—original draft, review, and editing.
J Kaneshiro: software and methodology.
M Takeda: resources, investigation, and methodology.
K Sasaki: validation, investigation, and methodology.
R Yamamoto: investigation.
D Umetsu: resources, investigation, and methodology.
E Kuranaga: resources, investigation, and methodology.
S Higo: resources.
T Kondo: resources.
Y Asano: resources.
Y Sakata: resources.
S Miyagawa: conceptualization, resources, and funding acquisition.
TM Watanabe: conceptualization, formal analysis, funding acquisition, project administration, and writing—original draft, review, and editing.

## Conflict of Interest Statement

The authors declare that they have no conflict of interest.

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
