## [Reviewer comments · Life Science Alliance]

Life Science Alliance

Estimation of crossbridge-state during cardiomyocyte beating using second harmonic generation

Tomonobu Watanabe, Hideaki Fujita, Junichi Kaneshiro, Maki Takeda, Kensuke Sasaki, Rikako Yamamoto, Daiki Umetsu, Erina Kuranaga, Shuichiro Higo, Takumi Kondo, Yoshihiro Asano, Yasushi Sakata, and Shigeru Miyagawa

DOI: <https://doi.org/10.26508/lsa.202302070>

Corresponding author(s): Tomonobu Watanabe, RIKEN Center for Biosystems Dynamics Research

Review Timeline:	Submission Date:	2023-04-02
	Editorial Decision:	2023-04-03
	Revision Received:	2023-04-03
	Editorial Decision:	2023-04-11
	Revision Received:	2023-04-21
	Accepted:	2023-04-21

Transaction Report:

Please note that the manuscript was previously reviewed at another journal and the reports were taken into account in the decision-making process at Life Science Alliance.

Referee #1 Review

Report for Author:

Here, the authors utilized second harmonic generation to assess myosin force generation in living contracting cardiomyocytes utilizing a previously published highly sensitive SHG microscope with a fast polarization-controllable device (Kaneshiro et al., 2019, Kaneshiro et al., 2016). Previously, this technology has already been utilized to analyze dissected intact frog tibialis anterior muscle fiber during isometric force generation (Nucciotti et al., 2010), worm walking in *Caenorhabditis elegans* (Psilodimitrakopoulos et al., 2014) and collagenase-treated interosseal cells (Forderer et al., 2016). In order to successfully analyze isolated cardiomyocytes, the authors utilized a line-and-space pattern substrate, which could solve the problem of sarcomere arrangement irregularity, as previously reported. To validate their approach, the authors utilized several drugs as well as disease-causing mutations which are known to affect cardiomyocyte contraction. In addition, they validated the system by inducing photodamage. Finally, they performed an intravital evaluation of force generation dysfunction in a *Drosophila* model of Barth syndrome.

Major comments:

- 1) The authors state that they had to solve two problems. One problem was the alignment of myofibrils in cardiomyocytes. This applies only to immature hiPSC-derived cardiomyocytes. Notably, this issue has been solved long time ago when it has been shown that patterned surfaces can be utilized to grow cardiomyocytes in lines resulting in organized myofibrils (as also stated by the authors). The second problem is the low temporal resolution of current SHG polarization measurements. Yet, it appears the used setup has been published previously.
- 2) The authors have focused mainly on very immature cardiomyocytes (hiPSC-derived, represent embryonic cardiomyocytes; *drosophila*). Authors should include in their measurements primary cardiomyocytes (rat neonatal and adult (need to be stimulated)) and provide a comparison. It would be further of great interest to apply the method to engineered cardiac tissues and compare it to native heart muscle.
- 3) The authors provide clear evidence that their method is applicable to analyze cardiomyocytes and the effect of drugs and mutations. Thus, it could be important to provide a statement how difficult it is to apply this technology. Is it practical?
- 4) This reviewer would assume that force generation depends on the mechanical properties of the substrate the cardiomyocytes adhere to. Please comment. Further, this reviewer would assume that force measurements would depend on whether the cells have to work against a force or not. Please comment.

Minor concerns:

- 1) The English needs to be improved.
- 2) The text is often difficult to understand.
- 3) Authors should better introduce the term "crossbridge-state". Could for example be explained in context to Figure 1A.
- 4) Discussion is very long
- 5) Figures are overloaded.

Referee #2 Review

Report for Author:

The manuscript "Estimation of crossbridge-state during cardiomyocyte beating using second harmonic generation" (SHG) by Fujita et al. presents an assay to measure SHG polarization dynamically in said samples. The authors claim that they have proof that the SHG anisotropy measured with their assay depend on the crossbridge status of the samples, yielding a method to evaluate myosin force generation in various samples, including ones with inheritable mutations or ones pretreated with UV light. In addition, the assay has been tested intravitaly in a *Drosophila* disease model.

These findings are new and significant and all conclusions are sufficiently supported by data. The manuscript may very well lay ground to subsequent research in this field, because the assay can be used for many different samples, as mentioned above.

The figures are designed straight forwarded and provide, in my opinion, no obstacle to understand the findings of the study. The results are thoroughly discussed in the context of previous research, but in the writing, the results and their discussion are often mixed together which makes reading difficult.

Yet overall, I'm pretty impressed with the quality of the study and the data. This is also true for the technical aspects of polarization dependent SHG microscopy which the authors have carried out with obvious care. However, it might be good to include the discussion of one specific aspect: The incident laser beam is reflected by three mirrors (two in the scanning system - are these silver coated? - and one dichroic) before it is focused down by the objective lens. Depending on the coating, reflections on mirrors can have subtle influences on the polarization state of the laser beam, introducing ellipticity, especially for polarization angles different from 0 and 90 degrees. This residual ellipticity might be small, posing no problems to the interpretation of the data, but its size should be estimated (or quantified, if possible, e.g., by installing a polarizer in the beam path after OL2 (cf. figure S2) and measuring the residual background level with a "null" sample) and its impact on the data interpretation should be discussed.

Referee #3 Review

Report for Author:

Fujita et al. use second harmonic generation to examine cardiomyocyte beating in several cardiac model systems. There are multiple issues with the manuscript that need to be addressed.

The significance of these studies is not clear. SHG has previously been used to study muscle. The interpretation of gamma as a measure of the crossbridge state has been established previously by multiple studies (and this point is extensively discussed in the discussion), and this is supported by previous experiments using ATP analogs. Moreover, SHG has been applied to study muscle across scales of organization, including muscle from *C. elegans*. The application of SHG to beating cardiomyocytes is interesting, but it is not clear how this significantly advances the field.

There are significant issues with the data and its interpretation. A non-exhaustive list of examples includes:

- Throughout the text, the authors point out where data deviates from their model and provide an explanation of why this might be, but do not provide evidence to back up these assertions. For example, in Fig. 4D, it is stated that gamma is normalized at high doses of radiation due to the emergence of a radiation-resistant population of cells (p. 15), but this is not tested. This is one of several examples.
- In Fig. 4B, it is stated that UV causes sarcomeric disarray, but the sarcomeres in the control cell look more disordered than the irradiated cells. Moreover, there are a large number of cells in the control where DAPI is visible, but there is no sarcomeric staining.
- In Fig. 2H, it is claimed that beating starts upon irradiation, but it is not clear how these peaks were identified (or whether these peaks are different from the unirradiated data).
- The authors discuss gamma as a measure of the super relaxed state in some places, but do not test this assertion and they do not consider this assertion in other places. For example, both OM and the Mybpc3 mutations should decrease the super relaxed state, but this is only considered in 1 case.
- Gamma and SL are clearly correlated (see Fig. 3F, 4E etc.) and it is not clear how these effects can be separated.

There are significant issues with the statistics. For each experiment, the number of technical and biological replicates should be provided. ANOVA and t-tests require normally distributed data, and in places where these tests were used here, the data are not normally distributed. P-values need to be adjusted for multiple comparisons. Moreover, the definition of significance needs to be clearly stated. For example, on p. 9 it is stated that $p=0.02$ is different while $p=0.03$ is not. Finally, the calculated p-values are unreasonably low given the data. For example, in Fig. 4D, the points clearly overlap between conditions, but the p-value is 10^{-5} . My guess is that this comes from how N is defined.

The text needs proofreading. For example, there is a sixth figure in the manuscript that has no caption and is not referred to in the text. Also, values reported in figures differ from values reported in the text (e.g., the peaks of the green histogram shown in Fig. 3B and the text value on p. 6).

April 3, 2023

Re: Life Science Alliance manuscript #LSA-2023-02070-T

Prof. Tomonobu M Watanabe
Laboratory for Comprehensive Bioimaging
6-2-3 Furuedai, Suita-shi
Osaka 565-0874
Japan

Dear Dr. Watanabe,

Thank you for submitting your manuscript entitled "Estimation of crossbridge-state during cardiomyocyte beating using second harmonic generation" to Life Science Alliance. We would like to invite further consideration of this manuscript pending the following revisions:

- Address Reviewer 1's comments, excluding major comment #2.
- Address Reviewer 2's comments.
- Address Reviewer 3's comments.

Thank you for this interesting contribution to Life Science Alliance. We are looking forward to receiving your revised manuscript.

Sincerely,

- A letter addressing the reviewers' comments point by point.
- An editable version of the final text (.DOC or .DOCX) is needed for copyediting (no PDFs).
- High-resolution figure, supplementary figure and video files uploaded as individual files: See our detailed guidelines for preparing your production-ready images, <https://www.life-science-alliance.org/authors>
- Summary blurb (enter in submission system): A short text summarizing in a single sentence the study (max. 200 characters including spaces). This text is used in conjunction with the titles of papers, hence should be informative and complementary to the title and running title. It should describe the context and significance of the findings for a general readership; it should be written in the present tense and refer to the work in the third person. Author names should not be mentioned.
- By submitting a revision, you attest that you are aware of our payment policies found here: <https://www.life-science-alliance.org/copyright-license-fee>

B. MANUSCRIPT ORGANIZATION AND FORMATTING:

Referee #1:

Major comments:

1) The authors state that they had to solve two problems. One problem was the alignment of myofibrils in cardiomyocytes. This applies only to immature hiPSC-derived cardiomyocytes. Notably, this issue has been solved long time ago when it has been shown that patterned surfaces can be utilized to grow cardiomyocytes in lines resulting in organized myofibrils (as also stated by the authors). The second problem is the low temporal resolution of current SHG polarization measurements. Yet, it appears the used setup has been published previously.

Previous report on muscle SHG anisotropy measurements target large muscle fiber or bundle of myofibril. In this study, we succeeded in SHG anisotropy measurement from single sarcomere in beating cardiomyocyte, which was not possible in the past. Novelty of our work is combining patterned substrate culture to align sarcomere in cardiomyocyte and fast SHG measurement, which was the key for the SHG anisotropy measurement of beating cardiomyocytes. This is now stated in the introduction and discussion sections of the revised manuscript.

2) The authors have focused mainly on very immature cardiomyocytes (hiPSC-derived, represent embryonic cardiomyocytes; drosophila). Authors should include in their measurements primary cardiomyocytes (rat neonatal and adult (need to be stimulated)) and provide a comparison. It would be further of great interest to apply the method to engineered cardiac tissues and compare it to native heart muscle.

Excluded for revision.

3) The authors provide clear evidence that their method is applicable to analyze cardiomyocytes and the effect of drugs and mutations. Thus, it could be important to provide a statement how difficult it is to apply this technology. Is it practical?

Although the microscope setup may need severe adjustment by professional optical scientist, measurement itself is simple and reproducible and analysis could be done by commercial software. Thus, we believe that the method presented in this manuscript is practical for analyzing the effect of drugs and mutations. This is now stated in the discussion section of the revised manuscript.

4) This reviewer would assume that force generation depends on the mechanical properties of the substrate the cardiomyocytes adhere to. Please comment. Further, this reviewer would assume that force measurements would depend on whether the cells have to work against a force or not. Please comment.

We agree with the reviewer #1 that force generation depends on the mechanical properties of the substrate. The stiffness of the substrate, which is glass in our case, is at mega Pascal range. On the other hand, sarcomeres shorten within the cardiomyocyte which is very soft compared to the substrate. Thus, sarcomere is influenced by the mechanical properties of the cardiomyocyte rather than the substrate.

In this study, we only speculate the population of force generating crossbridges from γ value and did not mention the actual force because it is hard to estimate the actual force. However, we believe that knowing the population of force generating crossbridges alone is valuable when evaluating the effect of drugs or genetic defect. This is now stated in the discussion section of the revised manuscript.

Minor concerns:

1) The English needs to be improved.

The final manuscript of the manuscript is checked by native English speaker.

2) The text is often difficult to understand.

We have rephrased some of the sentences for easier reading in the revised manuscript.

3) Authors should better introduce the term "crossbridge-state". Could for example be explained in context to Figure 1A.

Crossbridge state is now explained in Fig. 1A in the revised manuscript.

4) Discussion is very long

We have replaced some of the paragraph into the supplementary text for easier reading.

5) Figures are overloaded.

We have revised the figures and separated some panels in the different figures. Figure number is now increased to 7 in the revised manuscript.

Referee #2:

The manuscript "Estimation of crossbridge-state during cardiomyocyte beating using second harmonic generation" (SHG) by Fujita et al. presents an assay to measure SHG polarization dynamically in said samples. The authors claim that they have proof that the SHG anisotropy measured with their assay depend on the crossbridge status of the samples, yielding a method to

evaluate myosin force generation in various samples, including ones with inheritable mutations or ones pretreated with UV light. In addition, the assay has been tested intravitaly in a *Drosophila* disease model.

These findings are new and significant and all conclusions are sufficiently supported by data. The manuscript may very well lay ground to subsequent research in this field, because the assay can be used for many different samples, as mentioned above.

The figures are designed straight forwarded and provide, in my opinion, no obstacle to understand the findings of the study. The results are thoroughly discussed in the context of previous research, but in the writing, the results and their discussion are often mixed together which makes reading difficult.

Thank you very much for your comment. In the initial submission, we included our interpretation of the data in the results section, which need to be experimentally tested. We have deleted or moved our interpretation and hypothesis to the discussion section in the revised manuscript.

Yet overall, I'm pretty impressed with the quality of the study and the data. This is also true for the technical aspects of polarization dependent SHG microscopy which the authors have carried out with obvious care. However, it might be good to include the discussion of one specific aspect: The incident laser beam is reflected by three mirrors (two in the scanning system - are these silver coated? - and one dichroic) before it is focused down by the objective lens. Depending on the coating, reflections on mirrors can have subtle influences on the polarization state of the laser beam, introducing ellipticity, especially for polarization angles different from 0 and 90 degrees. This residual ellipticity might be small, posing no problems to the interpretation of the data, but its size should be estimated (or quantified, if possible, e.g., by installing a polarizer in the beam path after OL2 (cf. figure S2) and measuring the residual background level with a "null" sample) and its impact on the data interpretation should be discussed.

We would like to appreciate for the reviewer's valuable comment. For parallel applicability with fluorescent imaging, a laser scanning optical microscope system employed herein involves silver-coated mirrors, beam splitting mirrors, such as dielectric-multilayered dichroic mirrors and oxide-coated half-mirrors, sharing light paths, potentially causing polarization distortion, as the reviewer pointed out. We previously discussed this issue during the development of a high-speed polarization control system (Kaneshiro J, et al., *Appl Opt.* 2016 55:1082, doi: 10.1364/AO.55.001082). We conducted the calibration of polarization intensity with

estimation of the phase retardation parameters from experimental data in the “null” sample condition. The optical set up and the calibration procedure was described in the literature. This literature was already cited but we added short sentences about this into the Methods section.

Referee #3:

The significance of these studies is not clear. SHG has previously been used to study muscle. The interpretation of gamma as a measure of the crossbridge state has been established previously by multiple studies (and this point is extensively discussed in the discussion), and this is supported by previous experiments using ATP analogs. Moreover, SHG has been applied to study muscle across scales of organization, including muscle from *C. elegans*. The application of SHG to beating cardiomyocytes is interesting, but it is not clear how this significantly advances the field.

Failure in heart is critical but assessing the abnormality of cardiomyocyte before the appearance of the symptom is difficult. Although more than 200 genes are reported to be the causal gene for familial cardiomyopathy, it is difficult to predict cardiomyopathy by genomic inspection alone because cardiomyopathy can be caused by multiple gene mutations despite these mutation in single gene may not cause cardiomyopathy. Inspection of iPSC-derived cardiomyocyte may enable to foresee the cardiomyopathy of the individual before the appearance of the symptom. Indeed, electrophysiological inspection of iPSC-derived cardiomyocyte is previously reported. In this manuscript, we report the contractility inspection method of iPSC-derived cardiomyocytes. In this aspect, SHG observation of beating cardiomyocyte is essential, which we introduced in this manuscript. This is now stated in the abstract section of the revised manuscript.

There are significant issues with the data and its interpretation. A non-exhaustive list of examples includes:

-Throughout the text, the authors point out where data deviates from their model and provide an explanation of why this might be, but do not provide evidence to back up these assertions. For example, in Fig. 4D, it is stated that gamma is normalized at high doses of radiation due to the emergence of a radiation-resistant population of cells (p. 15), but this is not tested. This is one of several examples.

In the initial submission, we included our interpretation of the data in the results section, which need to be experimentally tested. We have moved or deleted our interpretation and hypothesis to the discussion section of the revised manuscript.

-In Fig. 4B, it is stated that UV causes sarcomeric disarray, but the sarcomeres in the control cell

look more disordered than the irradiated cells. Moreover, there are a large number of cells in the control where DAPI is visible, but there is no sarcomeric staining.

We agree with the reviewer that it is difficult to observe the sarcomeric disarray from the fluorescent images. Thus, we have deleted this statement in the revised manuscript.

Differentiation efficiency of hiPS to cardiomyocytes are not 100% and undifferentiated cells surround cardiomyocytes, which is typical with the protocol we used. This is now stated in the revised manuscript.

-In Fig. 2H, it is claimed that beating starts upon irradiation, but it is not clear how these peaks were identified (or whether these peaks are different from the unirradiated data).

Beating was confirmed by SHG video. This is now stated in the figure legend of the revised manuscript.

-The authors discuss gamma as a measure of the super relaxed state in some places, but do not test this assertion and they do not consider this assertion in other places. For example, both OM and the Mybpc3 mutations should decrease the super relaxed state, but this is only considered in 1 case.

The assumption that gamma may be a measure of super relaxed state is just a speculation and is not tested in this study, thus, should not appear in the results section. We have relocated the sentences related to super relaxation in the Discussion section of the revised manuscript.

-Gamma and SL are clearly correlated (see Fig. 3F, 4E etc.) and it is not clear how these effects can be separated.

There are clear correlations between gamma and SL. What is important for assessing the working crossbridge during contraction is the slope of the gamma-SL relationship. This is now stated in the revised manuscript.

There are significant issues with the statistics. For each experiment, the number of technical and biological replicates should be provided. ANOVA and t-tests require normally distributed data, and in places where these tests were used here, the data are not normally distributed. P-values need to be adjusted for multiple comparisons. Moreover, the definition of significance needs to be clearly stated. For example, on p. 9 it is stated that $p=0.02$ is different while $p=0.03$ is not. Finally, the calculated p-values are unreasonably low given the data. For example, in Fig. 4D, the points clearly overlap between conditions, but the p-value is 10^{-5} . My guess is that this comes from how N is defined.

We agree with the reviewer that in the cases where data are not normally distributed,

t-test shouldn't be used. In the revised manuscript, we used Mann-Whitney's U-test and revised the p values. Thank you very much for your comment.

The text needs proofreading. For example, there is a sixth figure in the manuscript that has no caption and is not referred to in the text. Also, values reported in figures differ from values reported in the text (e.g., the peaks of the green histogram shown in Fig. 3B and the text value on p. 6).

We apologized for our careless mistakes. We have corrected our mistakes and the final manuscript were checked by the English editing company in the revised manuscript.

April 11, 2023

RE: Life Science Alliance Manuscript #LSA-2023-02070-TR

Prof. Tomonobu M Watanabe
RIKEN Center for Biosystems Dynamics Research
2-2-3, Minatomachi-minami, Chuo-ku
Kobe 650-0047
Japan

Dear Dr. Watanabe,

Thank you for submitting your revised manuscript entitled "Estimation of crossbridge-state during cardiomyocyte beating using second harmonic generation". We would be happy to publish your paper in Life Science Alliance pending final revisions necessary to meet our formatting guidelines.

- please move the supplemental figure legends to the end of the figure legends list in the main text
- each supplemental figure should be uploaded as an individual file
- the supplemental discussion and associated references should be incorporated into the main document
- please add ORCID ID for corresponding author--you should have received instructions on how to do so

Figure Check:

- Figure S8 remains, but any mention of it in the text has been removed

A. FINAL FILES:

B. MANUSCRIPT ORGANIZATION AND FORMATTING:

Sincerely,

April 21, 2023

RE: Life Science Alliance Manuscript #LSA-2023-02070-TRR

Prof. Tomonobu M Watanabe
RIKEN Center for Biosystems Dynamics Research
2-2-3, Minatomachi-minami, Chuo-ku
Kobe 650-0047
Japan

Dear Dr. Watanabe,

Thank you for submitting your Methods entitled "Estimation of crossbridge-state during cardiomyocyte beating using second harmonic generation". It is a pleasure to let you know that your manuscript is now accepted for publication in Life Science Alliance. Congratulations on this interesting work.

DISTRIBUTION OF MATERIALS:

Again, congratulations on a very nice paper. I hope you found the review process to be constructive and are pleased with how the manuscript was handled editorially. We look forward to future exciting submissions from your lab.

Sincerely,
